# Fibroadipogenic progenitors are responsible for muscle loss in limb girdle muscular dystrophy 2B

Marshall W. Hogarth[1], Aurelia Defour[1], Christopher Lazarski[2], Eduard Gallardo[3,4], Jordi Diaz Manera[3,4], Terence A. Partridge[1,5], Kanneboyina Nagaraju[1,5,6] & Jyoti K. Jaiswal [1,5]

Muscle loss due to fibrotic or adipogenic replacement of myofibers is common in muscle diseases and muscle-resident fibro/adipogenic precursors (FAPs) are implicated in this process. While FAP-mediated muscle fibrosis is widely studied in muscle diseases, the role of FAPs in adipogenic muscle loss is not well understood. Adipogenic muscle loss is a feature of limb girdle muscular dystrophy 2B (LGMD2B) – a disease caused by mutations in dysferlin. Here we show that FAPs cause the adipogenic loss of dysferlin deficient muscle. Progressive accumulation of Annexin A2 (AnxA2) in the myofiber matrix causes FAP differentiation into adipocytes. Lack of AnxA2 prevents FAP adipogenesis, protecting against adipogenic loss of dysferlinopathic muscle while exogenous AnxA2 enhances muscle loss. Pharmacological inhibition of FAP adipogenesis arrests adipogenic replacement and degeneration of dysferlin-deficient muscle. These results demonstrate the pathogenic role of FAPs in LGMD2B and establish these cells as therapeutic targets to ameliorate muscle loss in patients.

[1] Research Center for Genetic Medicine, Children's Research Institute, Children's National Health System, Washington, DC 20010, USA. [2] Children's Research Institute, Children's National Health System, Washington, DC 20010, USA. [3] Unitat and Laboratori de Malaltdies Neuromusculars, Institut de Recerca, Hospital de la Santa Creu i Sant Pau, Universitat Autònoma de Barcelona, Barcelona 08041, Spain. [4] Centro de Investigación Biomédica en Red sobre Enfermedades Raras (CIBERER), Unitat de Malalties Neuromusculars, Servei de Neurologia, Hospital de la Santa Creu i Sant Pau de Barcelona, Barcelona 08041, Spain. [5] Department of Genomics and Precision Medicine, George Washington University School of Medicine, Washington, DC 20052, USA. [6] Department of Pharmaceutical Sciences, School of Pharmacy and Pharmaceutical Sciences, Binghamton University, Binghamton, NY 13902, USA. Correspondence and requests for materials should be addressed to J.K.J. (email: jkjaiswal@cnmc.org)

Although composed of terminally differentiated multi-nucleated myofibers, adult skeletal muscle maintains a remarkable ability to regenerate from injury. This ability depends on mono-nucleated cells that reside amongst the skeletal myofibers and those that enter the muscle following injury. With the ability of the satellite cells to proliferate and fuse to regenerate damaged myofibers, they have long been identified as the primary driver of regeneration. Accordingly, ablating the Pax7[+] satellite cells in adult mice blocks myofiber regeneration[1–3]. However, there is growing evidence that myofiber regeneration involves complex multicellular and extracellular matrix interactions creating a regenerative niche that consists of secreted factors, immune cells, myogenic and non-myogenic progenitors[4–7].

Fibro/adipogenic precursors (FAPs) are muscle-resident non-myogenic progenitors of mesenchymal origin that are marked by cell surface expression of platelet-derived growth factor receptor alpha (PDGFRα) and stem cell antigen-1 (Sca1) that proliferate in response to injury and can undergo fibrogenic or adipogenic differentiation[8–10]. Muscle injury triggers an acute myofiber repair response, failure of which causes myofiber death and resulting tissue infiltration by inflammatory cells[11–13]. These cells clear the debris from the injury site and activate both satellite cell and FAP proliferation[14,15]. A critical element in the regenerative process is transition of the pro-inflammatory cells to become pro-regenerative within 2–3 days after injury[16,17]. This coincides with the apoptotic clearance of FAPs and with satellite cell fusion leading to myogenesis. Timely occurrence of the above cellular choreography between inflammatory, fibro/adipogenic, and satellite cells has been implicated in successful muscle regeneration. Consequently, disrupting inflammatory infiltration and FAP homeostasis impairs regeneration, resulting in fibrotic and adipogenic degeneration of injured muscle[8,18]. This deficit has been demonstrated in the mouse model of Duchenne muscular dystrophy (mdx mice), where impaired FAP clearance results in muscle loss and fibrosis[19]. Facilitating apoptotic clearance of FAPs reduces muscle loss and improves mdx muscle function in vivo[19].

Adipogenic differentiation of FAPs has been implicated in muscle loss following rotator cuff injury in mice[20]. While adipogenic muscle replacement is prevalent in muscular dystrophies, it remains to be determined if FAPs are responsible for this. The dysferlinopathies represent a heterogeneous group of late-onset muscle disease, including limb girdle muscular dystrophy type 2B (LGMD2B), which are caused by mutations in the dysferlin gene[21–25]. Lack of dysferlin compromises myofiber repair, alters calcium homeostasis, and causes chronic muscle inflammation[21,26,27]. However, these deficits do not explain the late and abrupt disease onset, progressive nature, or specific muscle involvement seen in patients or mouse models. Recently it has been demonstrated that affected muscles of dysferlinopathic patients and mouse model show adipogenic replacement[28]. Unlike the myofiber repair deficit and inflammation, adipogenic replacement is observed only in symptomatic patient and mouse muscle[28,29]. Further, eccentric exercise exacerbates this phenotype in patients[30], suggesting a link between myofiber injury and adipogenic replacement of LGMD2B muscle. Muscle damage and disease severity in LGMD2B patients correlate with increased expression of another membrane repair protein Annexin A2 (AnxA2)[31,32]. Recently we described that dysferlin-deficient mice lacking AnxA2 have reduced myofiber repair ability, but are surprisingly protected from adipogenic myofiber loss[33]. This suggested that loss of AnxA2 disrupts the link between injury and adipogenic replacement of dysferlin-deficient myofibers.

Here, we study the effect of dysferlin loss on the homeostasis of muscle-resident FAPs. We examine if altered FAP biology can explain the late onset muscle-specific symptoms in LGMD2B

and if AnxA2 accumulation is a mediator of this process. By using dysferlinopathic patient and mouse models, we show that FAP accumulation and their adipogenic differentiation are key contributors to this muscular dystrophy. Importantly, the presence of extracellular AnxA2 promotes FAP proliferation and adipogenic differentiation, and the loss of AnxA2 or pharmacologically inhibiting FAP adipogenesis significantly ameliorates the dysferlin-deficient muscle pathology. This work identifies FAPs and their adipogenic differentiation as a major contributor to dysferlin-deficient muscle loss. By identifying approaches to target FAP proliferation and adipogenic differentiation, we provide novel therapeutic targets for treating LGMD2B.

## Results

**Muscle adipogenesis determines LGMD2B onset and severity**. MRI and histological analyses have identified fatty replacement of muscle in symptomatic dysferlinopathic patients[34] and mouse models[29,35]. By direct histological analysis of muscle sections from LGMD2B patients and mouse model, we examined how this association relates to disease severity. We obtained muscle biopsies from LGMD2B patients who exhibited mild to severe clinical symptoms described in Supplementary Table 1. As a first step, we used the neutral lipid stain Oil Red O to score the adipogenic status of muscle sections from these patients. While sections of healthy muscle showed little to no oil red staining, extensive staining was noted between the myofibers in symptomatic patient muscle sections, which increased with the severity of the patient's clinical symptoms and the extent of muscle loss (Fig. 1a and Supplementary Fig. 1). To further examine if the adipogenic deposits were originating from within the myofibers we examined the localization of Perilipin-1, a protein that coats the adipocyte lipid droplets[36]. No perilipin-1 staining was detectable in healthy muscle, but patient muscles showed extensive perilipin-1 labeling, which increased with disease severity and localized exclusively in the extracellular matrix space between the myofibers (Fig. 1b, c). Patient myofibers did not show internal perilipin-1 labeling even when they are adjacent to lipid deposits (Fig. 1c and Supplementary Fig. 1), suggesting extra-myofiber origin of these lipids.

To independently assess the link between disease severity and the extent of adipogenic replacement of muscle, we examined muscles from 12-month old dysferlin-deficient (B6A/J) mice (Fig. 1d). Disease severity, as indicated by the extent of damage and regeneration (myofiber central nucleation) showed a progressive increase between muscles in the following order: TA, gastrocnemius, quadriceps, psoas (Fig. 1e). Labeling with perilipin-1 showed a parallel increase in lipid accumulation between the myofibers across these same muscles (Fig. 1d, f). Similarly, muscle (gastrocnemius) collected from mice with increasing age (6–18 months old) showed progressively increasing adipogenic replacement marked by increased labeling with either Oil Red O (Fig. 1i) or perilipin-1 (Fig. 1j). Interestingly, while increased perilipin-1 staining is detected starting 12 months (Fig. 1h), these muscles showed an increased central nucleation starting from 6 months (Fig. 1g), suggesting that myofiber damage and regeneration precedes their later adipogenic replacement. Again, in the mouse muscle we observed perilipin-1 labeling only in the extracellular matrix and not in adjacent myofibers (Fig. 1k), suggesting that the lipid does not originate in muscle fibers, and instead is produced by muscle interstitial cells.

**FAPs cause the adipogenic loss of dysferlinopathic muscle**. The above results suggest the extent of muscle damage and regeneration preempts the degree of adipogenic replacement of dysferlinopathic muscle. With the known proliferation of FAPs

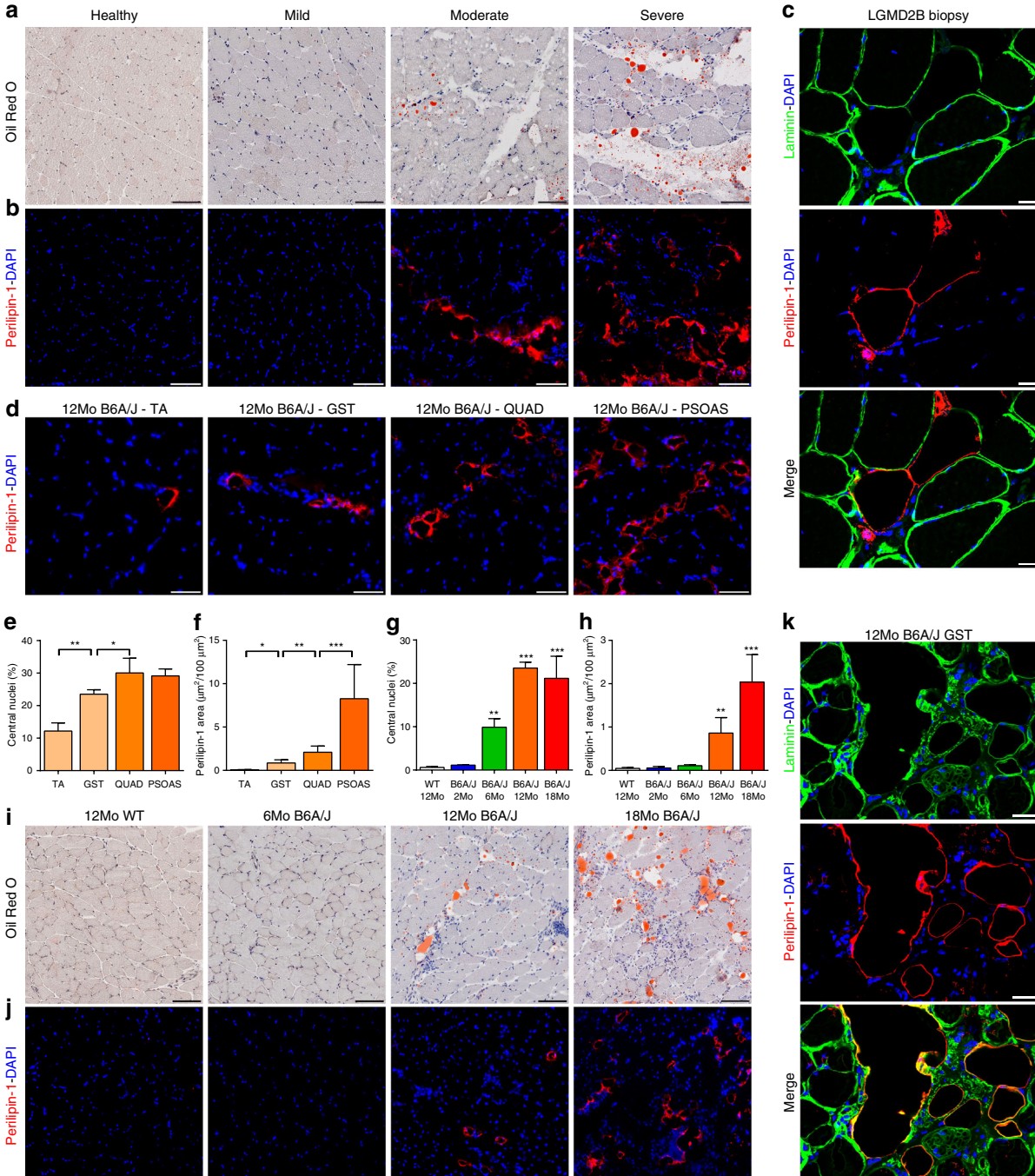

**Fig. 1** Adipogenic replacement of muscle correlates with disease severity in dysferlinopathic patients and mice. Mild, moderate, and severely dystrophic LGMD2B patient and non-dystrophic control muscle cross-section stained with **a** Oil Red O or for **b** Perilipin-1 protein. **c** Confocal images of LGMD2B patient muscle sections showing that perilipin-1 marked lipid deposits (red) accumulate outside the boundaries of laminin-marked myofiber borders (green). Scale bar = 20 μm. **d** 12Mo B6A/J TA, gastrocnemius, quadriceps, and psoas muscles stained for perilipin-1. Scale Bar = 100 μm. Quantification (mean ± SD) of **e** myofiber central nucleation and **f** perilipin-1 area across 12Mo B6A/J muscles. Statistical comparisons performed via *t*-test between adjacent groups, *n* = 4 muscles/group. Quantification (mean ± SD) of **g** myofiber central nucleation and **h** perilipin-1 area from B6A/J gastrocnemius with advancing age/pathology, *n* = 4 mice/group. Statistical comparisons performed via ANOVA with Holm–Sidak multiple comparisons test for all means with that of 12Mo WT, \**p* < 0.05 \*\**p* < 0.01, \*\*\**p* < 0.001. **i** Oil Red O and **j** Perilipin-1 labeling of gastrocnemius from 6, 12, and 18Mo B6A/J and 12Mo WT. Scale bar = 100 μm. **k** Confocal image of gastrocnemius muscle sections showing that perilipin-1 marked lipid deposits (red) localize outside the boundaries of laminin-marked myofiber borders (green) in 12Mo B6A/J. Scale bar = 20 μm

in the muscle interstitium following injury, and their adipogenic potential, we examined if FAPs lead to adipogenic conversion of LGMD2B muscle. Using PDGFRα to label FAPs, control biopsies showed minimal interstitial PDGFRα staining, which was evident in LGMD2B patients and increased with worsening clinical

severity (Fig. 2a, d). Analysis of mouse muscles showed a similar interstitial accumulation of PDGFRα in B6A/J, which correlates both to extent of disease severity across muscles at the same age (Fig. 2b, e), and to increasing disease severity due to age of the muscle (gastrocnemius) from 6 months onwards (Fig. 2c, f). This

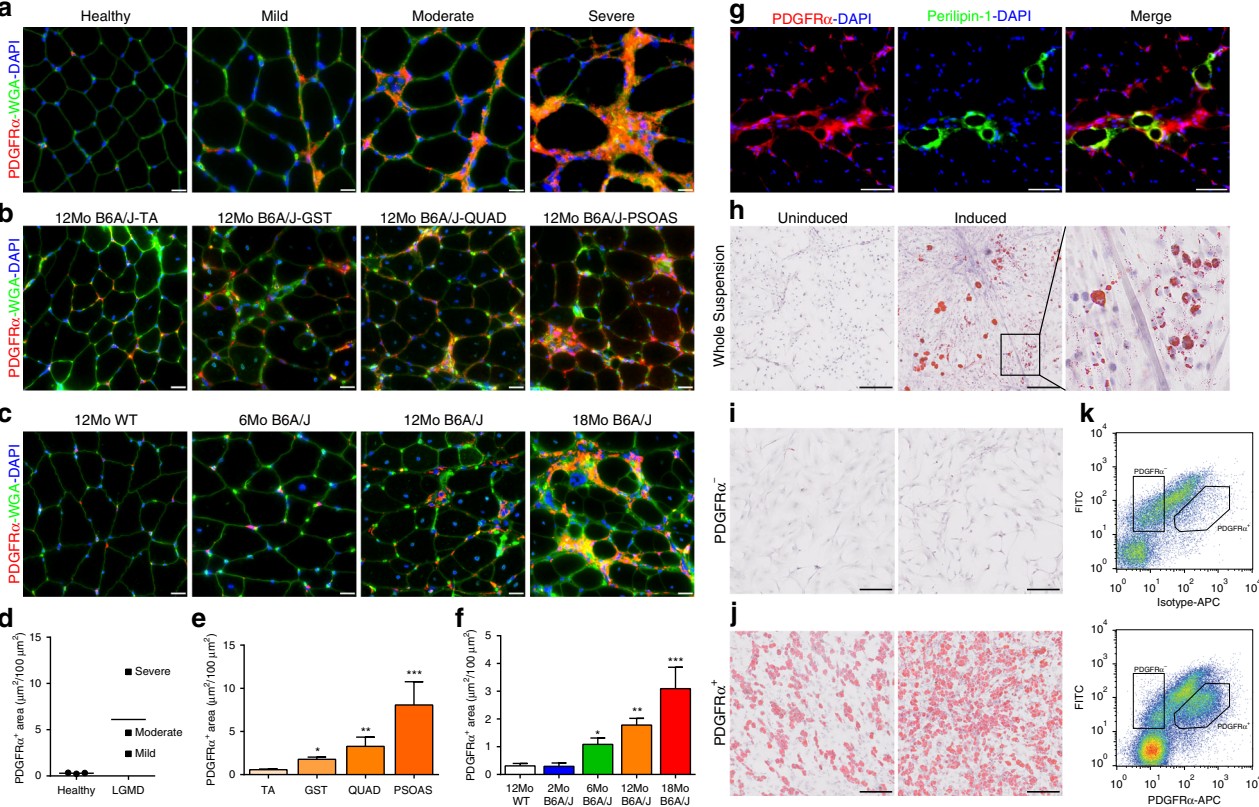

**Fig. 2** FAP accumulation and differentiation dictates the extent of adipogenic replacement of dysferlin-deficient muscle. PDGFRα staining of FAPs in **a** mild, moderate, and severe LGMD2B patient and non-dystrophic control muscle cross-sections; **b** muscles with increasing pathology from 12Mo B6A/J and **c** gastrocnemius muscle cross-sections from 6, 12, and 18Mo B6A/J and 12Mo WT control. Scale bar = 20 μm. Quantification (mean ± SD) of muscle area labeled with PDGFRα in **d** human and mouse muscle with **e** increasing pathology and **f** increasing age. $n = 4$ mice/group. Statistical comparisons performed via ANOVA with Holm–Sidak multiple comparisons test for all means with 12Mo TA or 12Mo WT respectively, *$p < 0.05$, **$p < 0.01$, ***$p < 0.001$. **g** Gastrocnemius muscle cross-section of 12Mo B6A/J co-labeled for PDGFRα and Perilipin-1. Scale bar = 50 μm. **h** Oil Red O staining of primary cell suspension isolated from hindlimb muscle of 6Mo B6A/J prior to (uninduced) or following (induced) induction for adipogenic differentiation. Scale bar = 200 μm. Inset shows a zoomed image of the box drawn in **h**. Oil Red O staining of sorted cells **i** lacking or **j** expressing PDGFRα at their cell surface. Scale bar = 200 μm. **k** FACS scatter plots showing cell surface labeling of PDGFRα in primary cell suspension prepared from mouse hindlimb muscle. The regions drawn mark the cells that express or lack cell surface PDGFRα expression

is a feature of the dysferlinopathic mice, and is not observed in the healthy, wild type (WT) mice (Supplementary Fig. 2). PDGFRα-labeled FAP increase occurred concomitantly with the increase in central nucleation, but prior to adipogenic replacement of the myofibers (Fig. 2f compared to Fig. 1f). This observation suggests that persistent myofiber injury and regeneration causes FAP accumulation. Subsequent adipogenic differentiation of these FAPs likely causes adipogenic replacement of the myofibers, as indicated by the formation of lipid deposits in extracellular matrix regions being enriched for PDGFRα-marked FAPs (Fig. 2g).

To directly assess the role of FAPs in adipogenic replacement of dysferlinopathic myofibers, we obtained a mixed primary cell suspension from hindlimb muscles of 6Mo B6A/J mice. Inducing cells with adipogenic media caused a proportion of cells to differentiate into oil red-labeled adipocytes, such that even when these adipocytes were found adjacent to myotubes, the myotubes themselves were not producing the lipids (Fig. 2h, and inset). This further supports that non-myogenic cells produce the adipogenic material that eventually replace myofibers. To establish the identity of these adipogenic cells we used fluorescence-activated cell sorting to isolate PDGFRα-labeled (PDGFRα+) FAPs and PDGFRα-unlabeled (PDGFRα−) cells from a primary muscle cell suspension (Fig. 2k). Even upon adipogenic induction, no adipocyte formation was observed in PDGFRα− cells (Fig. 2i).

However, PDGFRα+ cells formed adipocytes spontaneously, which increased further upon adipogenic induction (Fig. 2j). These results identify FAPs as the muscle interstitial cells that contribute to the adipogenic loss of dysferlin-deficient muscle.

**Adipogenic muscle loss depends on age and myofiber injury.** Based on above observations we hypothesized that age, chronic injury, and poor repair of dysferlinopathic myofibers create a niche that promotes progressive interstitial FAP accumulation and subsequent differentiation into adipocytes. To test this, we examined if injury and age of the TA (a muscle that is minimally affected in B6A/J) can unmask the adipogenic potential of the FAPs and effect of tissue injury and age on this process. TA muscles from 3Mo and 12Mo B6A/J mice were injured by notexin and after allowing 4 weeks for myofibers to fully regenerate, the extent of adipogenic muscle replacement was scored and compared to the uninjured 12mo B6A/J TA (Supplementary Fig. 3). The uninjured 12Mo, and the injured 3Mo B6A/J TAs both lack significant adipogenic replacement (assessed by perilipin-1 staining) (Fig. 3a–c). In contrast, the injured 12Mo B6A/J TA showed substantially increased adipogenic foci and myofiber areas replaced by adipogenesis (Fig. 3a–c). This showed that increasing age increases injury-triggered adipogenic replacement of areas otherwise occupied by myofibers. This link is

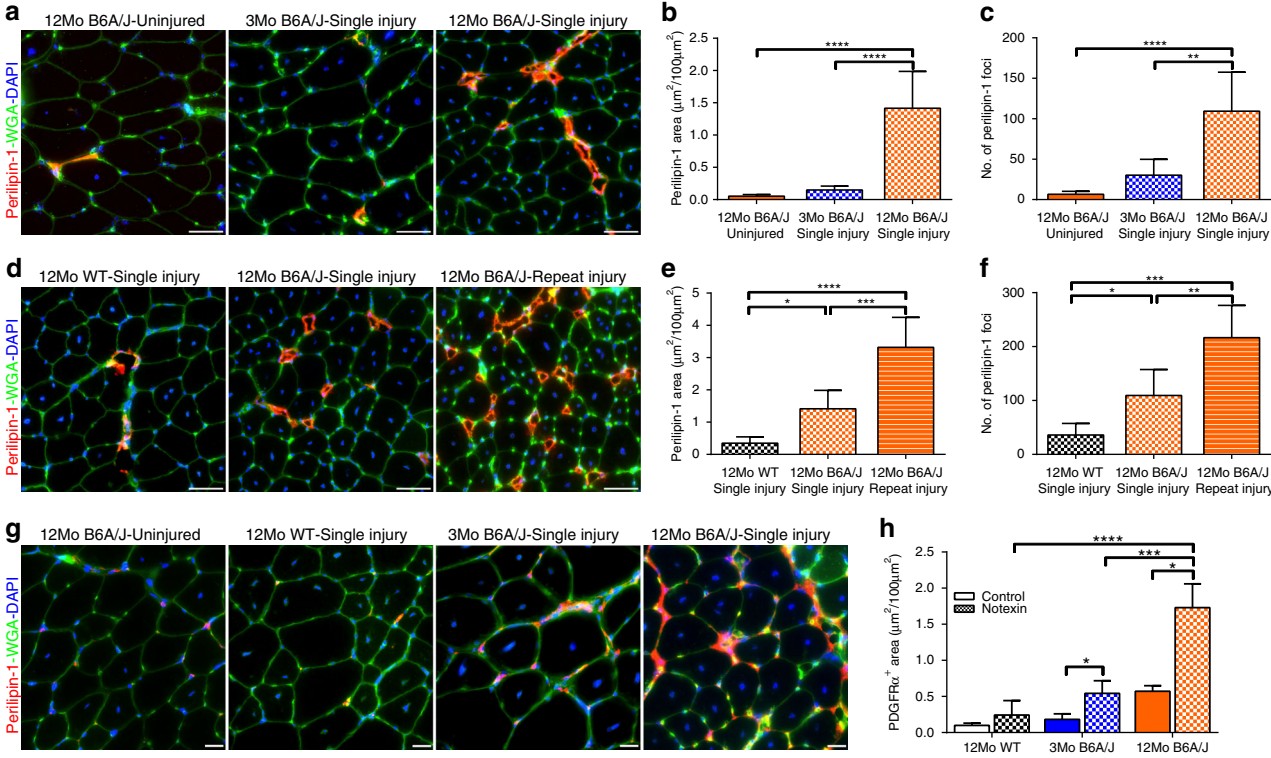

**Fig. 3** Injury induces FAP proliferation in older but not younger dysferlinopathic muscle. Images showing Perilipin-1 staining in cross-section of TA muscles from 2 and 12Mo in B6A/J or WT Bl6 mice following 4 weeks after **a** a single or **d** three rounds of notexin injury. Scale bar = 50 μm. Quantification (mean ± SD) in TA muscle cross-section of **b**, **e** total area and **c**, **f** number of foci labeled with perilipin-1 after 4 weeks of **b**, **c** single or **e**, **f** 3-rounds of notexin injury. n = 6 per group. Statistical comparisons performed via ANOVA with Holm–Sidak multiple comparisons test to compare all means with Uninjured 12Mo B6A/J, *p < 0.05, **p < 0.01, ***p < 0.001, ****p < 0.0001. **g** Images showing PDGFRα staining after single notexin injury of TA muscle from 12Mo WT, 2Mo and 12Mo B6A/J. Scale bar = 20 μm. **h** Quantification (mean ± SD) of PDGFRα-labeled area of the muscle after notexin injury, n = 6 mice/group. Statistical comparisons between control and notexin injured performed via two-tailed t-test for each group. Comparisons between notexin injured muscles is by ANOVA with Holm–Sidak multiple comparisons test to compare all means with 12Mo WT, *p < 0.05, ***p < 0.001, ****p < 0.0001

further strengthened by the observation that adipogenic replacement occurs at the site of notexin injection (marked by tattoo dye), while adjacent uninjured regions remained non-adipogenic (Supplementary Fig. 3). These results establish muscle injury and age enhance adipogenic replacement of dysferlinopathic muscle.

To confirm that this post-injury lipid formation was due to the absence of dysferlin, we also injured TAs from 12Mo WT, which showed no significant lipid formation after injury (Fig. 3d–f). Dysferlinopathic muscle undergoes repeat bouts of injury over the disease course, which we hypothesize leads to adipogenic replacement of the muscle. To test this, we performed three consecutive injuries to the TA muscle in dysferlinopathic mice, allowing 2-week regeneration periods between the injuries (Supplementary Fig. 3). This resulted in more adipogenic replacement of the TA muscle than in the single injured TA muscle (Fig. 3d–f), suggesting that repeat rounds of myofiber injury and regeneration drive the adipogenic conversion of dysferlinopathic muscle. To examine if this injury-dependent adipogenic replacement was a consequence of increased FAP accumulation caused by the injuries, we stained these muscles for PDGFRα (Fig. 3g). Compared to uninjured B6A/J muscles and WT muscles, there was a significant increase in PDGFRα-labeled FAPs in injured B6A/J muscles (Fig. 3h). Injured 12Mo B6A/J muscle accumulates significantly more FAPs than both the 3Mo B6A/J and the 12Mo WT injured muscles, indicating that excessive FAP accumulation after injury is a feature of dysferlin-deficient muscle with advancing pathology, and likely underpins the adipogenic loss of these muscles.

**AnxA2 links muscle injury to FAP activation and adipogenesis**. The above link between FAP accumulation and adipogenic replacement of dysferlinopathic muscle suggests that accumulation and adipogenic differentiation of FAPs is responsible for the decline in dysferlinopathic muscle function, and reversing this could provide a therapy for LGMD2B. AnxA2 is a membrane repair protein that is elevated upon muscle injury and increases in LGMD2B patient muscle in a manner that correlates with disease severity[31,32]. Previously, we showed that deletion of AnxA2 in dysferlinopathic muscle reduces adipogenic replacement and improves muscle function despite no improvement in myofiber repair[33]. Thus, we examined if AnxA2 contributes to FAP proliferation/adipogenic differentiation in dysferlinopathic muscle. Immunostaining for AnxA2 showed that in dysferlinopathic mice AnxA2 level increases with disease severity and that this increase is in the level of AnxA2 in the interstitium (Fig. 4a, b, e). Co-labeling with PDGFRα showed that the FAPs are enriched in the regions with accumulation of AnxA2 (Fig. 4c, d). AnxA2 expression increases after muscle injury, and extracellular AnxA2 can activate inflammation via toll-like receptor 4 (TLR4)[37,38]. We thus investigated if the inflammatory cells are enriched at the sites of AnxA2 accumulation, and found this to be the case; F4/80 marked macrophages accumulated in interstitial regions enriched for AnxA2 (Fig. 4f), and are located adjacent to the PDGFRα-labeled FAPs (Fig. 4g). To analyze if AnxA2 accumulation is causally linked to adipogenic replacement via FAPs we examined adipogenic replacement and FAP accumulation

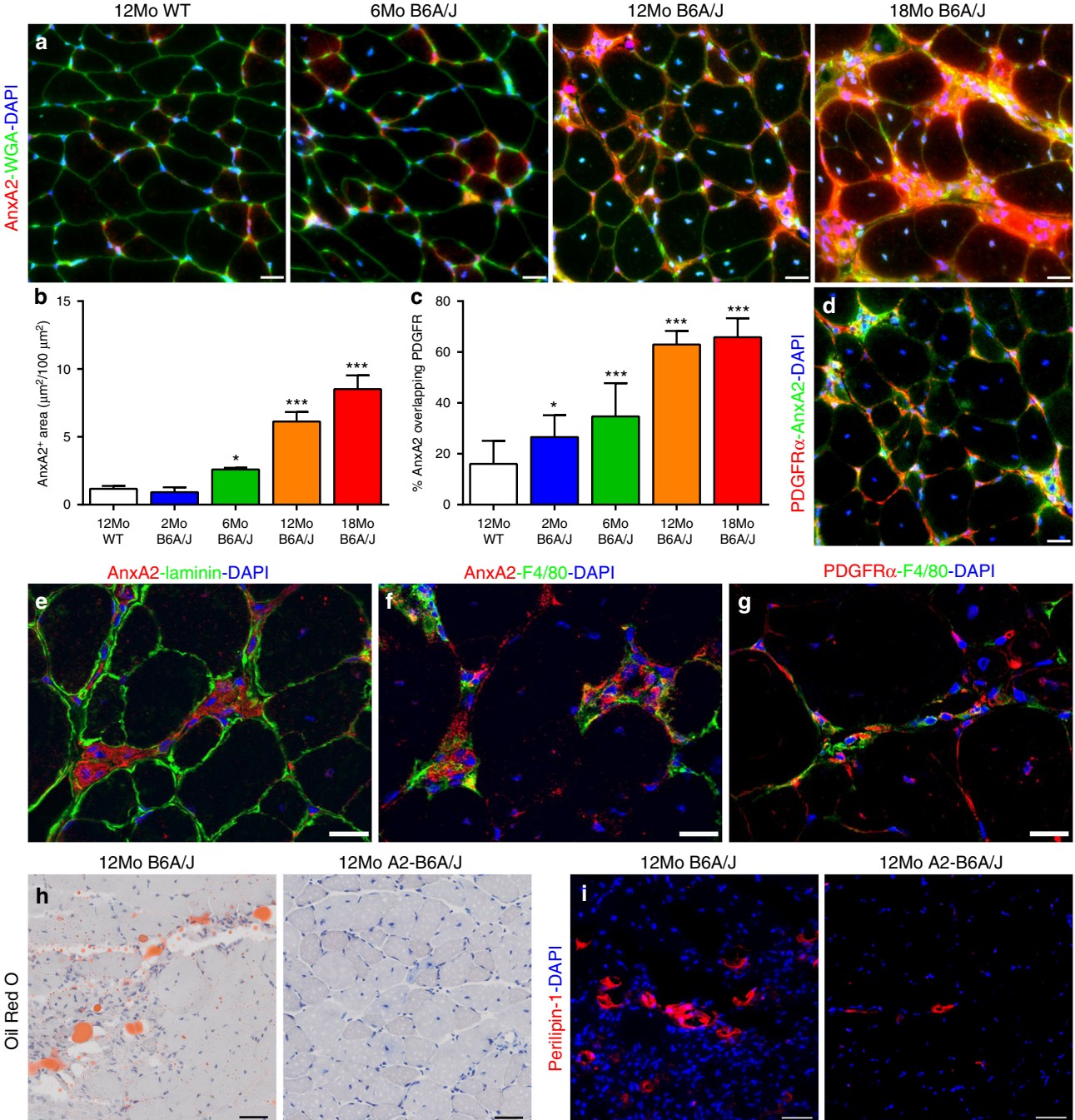

**Fig. 4** Extracellular accumulation of AnxA2 drives FAP accumulation in dysferlinopathic muscle. **a** AnxA2 staining in cross-sections from 6, 12, and 18Mo B6A/J and 12Mo WT mouse gastrocnemius muscle. Scale bar = 20 μm. **b** Quantification (mean ± SD) of total AnxA2-labeled area across the entire muscle cross-section, n = 4 mice/group. Statistical comparisons performed via ANOVA with Holm–Sidak multiple comparisons test to compare all means with 12Mo WT, *$p < 0.05$, **$p < 0.01$, ***$p < 0.001$. **c** Quantification (mean ± SD) of **d** co-localization between AnxA2 and PDGFRα in 12Mo B6A/J gastrocnemius, n = 4 mice/group. Statistical comparisons performed via ANOVA with Holm–Sidak multiple comparisons test to compare all means with Uninjured 12Mo WT, *$p < 0.05$, ***$p < 0.001$. Confocal images from 12Mo B6A/J gastrocnemius co-labeled for **e** AnxA2 and laminin, **f** AnxA2 and F4/80, and **g** PDGFRα and F4/80. Scale bar = 20 μm. **h** Oil Red O and **i** Perilipin-1 labeling of gastrocnemius cross-sections from 12Mo B6A/J and A2-B6A/J. Scale bar = 50 μm

in mice lacking dysferlin and AnxA2 (A2-B6A/J). Oil Red O and perilipin-1 labeling both reveal significantly less adipogenic replacement of muscle in A2-B6A/J than B6A/J mice (Fig. 4h, i, Supplementary Fig. 4 and quantified in Fig. 5d). This indicates that extracellular AnxA2 accumulation contributes to the pro-adipogenic niche as AnxA2 deletion arrests the adipogenic conversion of dysferlinopathic muscle.

As we find macrophages are enriched at sites of PDGFRα and AnxA2 accumulation, we next examined if the lack of extracellular AnxA2 works by inhibiting muscle inflammatory response. F4/80 staining of 12Mo gastrocnemius muscles showed that compared to the WT mice, both B6A/J and A2-B6A/J mice show a robust increase in macrophage infiltration (Fig. 5a, b). The extent of macrophage infiltration in these mouse muscles

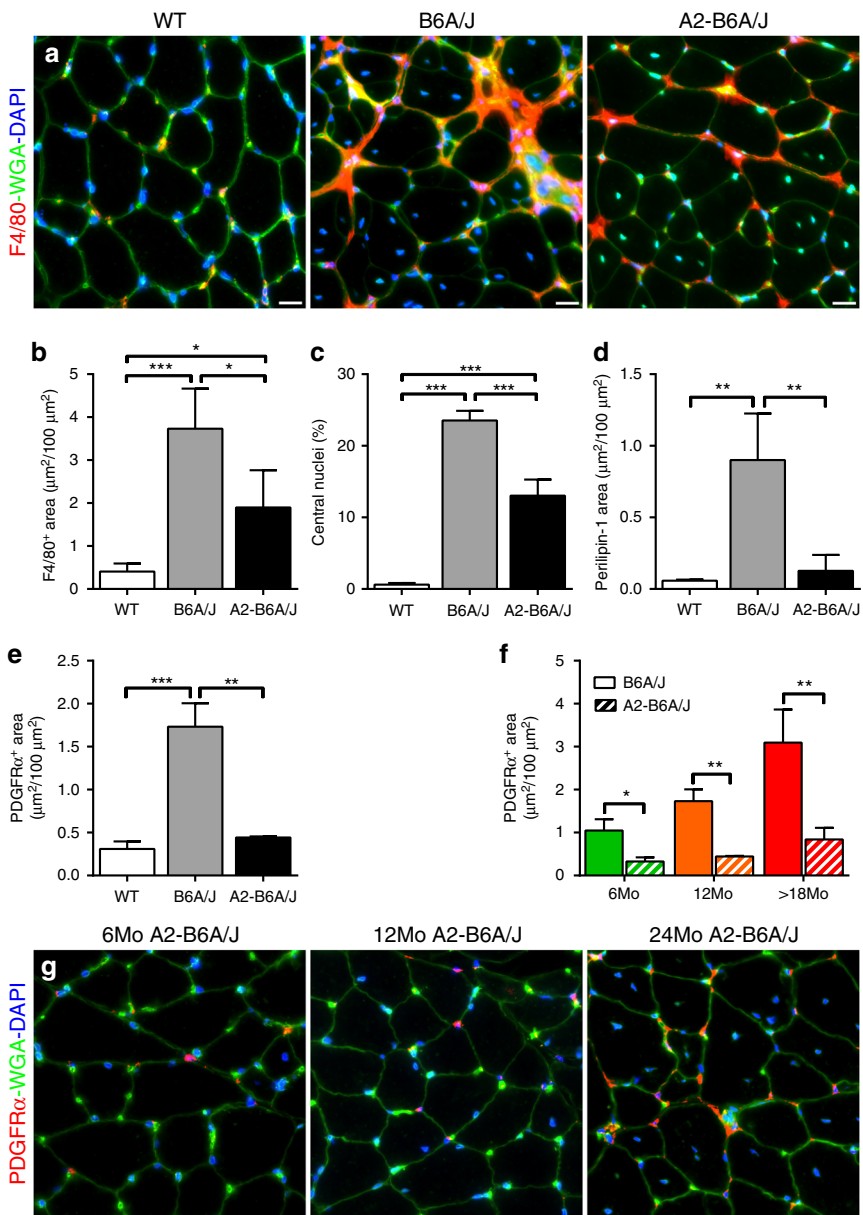

**Fig. 5** Presence of AnxA2 is required for adipogenic conversion of dysferlinopathic muscle. **a** F4/80 staining of gastrocnemius of 12Mo WT, B6A/J and A2-B6A/J. Scale bar = 20 µm. Quantification (mean ± SD) of **b** F4/80 labeled area, **c** myofiber central nucleation, **d** Perilipin-1 labeled area, and **e** PDGFRα labeled area from 12Mo WT, B6A/J and A2-B6A/J, $n = 4$/genotype. Statistical comparisons performed via ANOVA with Holm–Sidak multiple comparisons test to compare all means, $*p < 0.05$, $**p < 0.01$, $***p < 0.001$. **f** Quantification (mean ± SD) of **g** PDGFRα-labeled area in 6, 12, and 24Mo A2-B6A/J gastrocnemius (scale bar = 20 µm) shown in comparison to B6A/J (as presented in Fig. 2f), $n = 3$ mice/group. Statistical comparisons performed by two-tailed $t$-test between B6A/J and A2-B6A/J at each timepoint, $*p < 0.05$, $**p < 0.01$

is in line with the extent of myofiber injury, indicated by the number of centrally nucleated myofibers in B6A/J and A2-B6A/J mice as compared to WT mice (Fig. 5c). In contrast to the higher injury and inflammation of A2-B6A/J muscle (as compared to the WT), FAP (PDGFRα) accumulation and lipid (perilipin-1) formation were comparable between the A2-B6A/J and WT muscle (Fig. 5d, e). This raises the possibility that the beneficial effect of the lack of AnxA2 may be by way of its effect on FAP proliferation and adipogenic differentiation. To examine the effect of AnxA2 on FAP accumulation and on preventing their adipogenic differentiation, we first quantified FAP accumulation using PDGFRα labeling. While B6A/J mice showed increasing accumulation of FAPs with age, muscles in the A2-B6A/J mice showed fewer FAPs, and their numbers

did not increase with age (Fig. 5f, g). This indicates that lack of AnxA2 prevents FAP accumulation, contributing to the suppression of adipogenic replacement of dysferlin-deficient muscle.

We next examined if AnxA2 can also contribute to the adipogenic fate of the FAPs in dysferlin-deficient muscle. For this we isolated FAPs from 12Mo WT, B6A/J and A2-B6A/J muscle and quantified their spontaneous adipogenesis in vitro. A small proportion of WT FAPs undergo spontaneous adipogenesis after 14 days in culture, but B6A/J FAPs show significantly higher rates of adipogenesis (Fig. 6a, b). The spontaneous adipogenesis of B6A/J FAPs suggests these cells are committed to adipogenesis prior to extraction, which may be caused by the pro-adipogenic niche in dysferlinopathic muscle. The lack of spontaneous

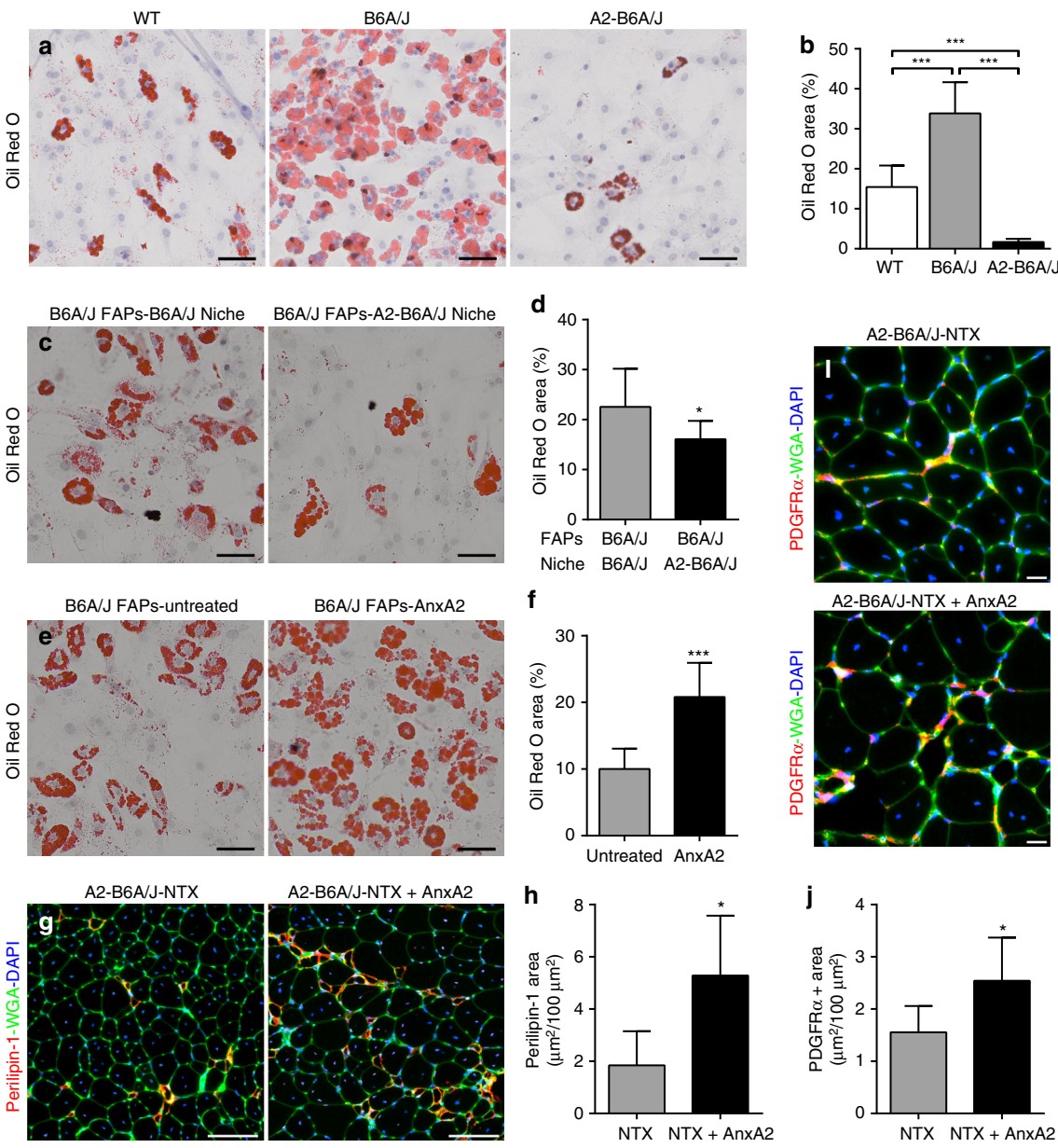

**Fig. 6** AnxA2 drives FAP adipogenesis and injury-triggered lipid formation in dysferlinopathic muscle. **a** Oil Red O staining and **b** quantification of spontaneous adipogenesis of 20,000 FAPs each, isolated from 12Mo WT, B6A/J and A2-B6A/J after 10 days in culture, $n = 3$ replicates/genotype. Scale bar = 50 μm. **c** Oil Red O staining and **d** quantification (normalized to 20,000 FAPs) of spontaneous adipogenesis from a 1:1 mixed culture of 24Mo B6A/J FAPs (10,000 cells) with niche cells (10,000 cells) from either 24Mo B6A/J or A2-B6A/J. Scale bar = 50 μm. **e** Oil Red O staining and **f** quantification of spontaneous adipogenesis of 20,000 24Mo B6A/J FAPs left untreated or treated with 100 nM recombinant AnxA2. Scale bar = 50 μm. **g** Perilipin-1 staining from 10Mo A2-B6A/J TA muscles 28d after single notexin injury with (NTX + AnxA2) or without (NTX) co-administration of 10 μg recombinant AnxA2. Scale bar = 100 μm. **h** Quantification of total perilipin-1 area after injury. **i** PDGFRα staining from 10Mo A2-B6A/J TA muscles 28d after single notexin injury and administration of 10 μg recombinant AnxA2 compared to notexin injury alone. Scale bar = 20 μm. **j** Quantification of total PDGFRα area after injury. All data presented as mean ± SD, average Oil Red area compared across genotypes by ANOVA with Holm–Sidak multiple comparisons test, all other data compared via two-tailed $t$-test, *$p \leq 0.05$ ***$p < 0.001$

adipogenesis in A2-B6A/J FAPs could be caused by the lack of a pro-adipogenic niche or restricted adipogenic potential of the FAPs in the absence of AnxA2 (Supplementary Fig. 4). As adipogenic replacement of muscle is enhanced with age, to allow for the niche cells to reach their maximal potential we isolated the niche (PDGFRα⁻) and FAP (PDGFRα⁺) cells from 24Mo B6A/J and A2-B6A/J muscles. These FAP and niche cells were then mixed in a 1:1 ratio (20,000 cells total; 10,000 FAPs) from the same or different genetic background and their spontaneous adipogenesis was quantified as the extent of oil red

staining/20,000 FAPs. Compared to the B6A/J niche, the A2-B6A/J niche restricted the spontaneous adipogenesis of B6A/J FAPs (Fig. 6c, d). Thus, the absence of AnxA2 in the niche cells reduces the adipogenicity of the dysferlinopathic FAPs. To directly test if it is the AnxA2 or another factor secreted by the niche cells that influences FAP adipogenesis we treated a purified population of B6A/J FAPs (20,0000 FAPs) with 100 nM AnxA2 and their spontaneous adipogenesis was quantified as the extent of oil red staining/20,000 FAPs. Compared to the AnxA2 untreated FAPs, treatment with purified AnxA2 alone caused a

>2-fold increase in the spontaneous adipogenic differentiation of these dysferlinopathic FAPs (Fig. 6e, f).

Taken together, the above results show that AnxA2 produced by the B6A/J muscle niche cells can potentiate the adipogenic differentiation of B6A/J FAPs in vitro. We next examined if this was also true in vivo. For this we used the AnxA2-naive A2-B6A/J mice and notexin injured their muscle with or without the addition of 10 μg purified AnxA2 at the site of injury. The presence of exogenous AnxA2 at the site of muscle regeneration in these otherwise AnxA2-deficient muscles resulted in increased accumulation of the PDGFR+ FAPs as well as perilipin-1 labeled adipogenic deposits (Fig. 6g–j). This provides direct evidence supporting that extracellular AnxA2 is not only necessary, but also sufficient for driving FAP-mediated in vivo adipogenic conversion of the regenerating dysferlinopathic muscles.

**Blocking FAP differentiation arrests adipogenic muscle loss.** Our data from the A2-B6A/J model suggests that restricting the adipogenic conversion would preserve the dysferlinopathic muscle and is thus a potential therapeutic target. To test this, and to independently confirm the benefit of inhibiting FAP adipogenesis for dysferlinopathy, we used a drug to inhibit the

adipogenic differentiation of FAPs. Batimastat is a small molecule drug that has previously been shown to restrict adipogenesis of both cultured adipogenic precursors[39] and WT mouse FAPs[40]. To test the ability of batimastat to prevent the spontaneous adipogenesis of B6A/J FAPs we treated 40,000 FAPs isolated from 12Mo B6A/J mice with 10 μM batimastat starting from 3 days in culture. Using Oil Red O, we quantified the extent of adipogenic differentiation of these FAPs after 14 days in culture in vitro (Fig. 7a). We find that treatment with batimastat resulted in reduced spontaneous adipogenesis of the B6A/J FAPs (Fig. 7b).

We next tested whether this effect of batimastat to repress FAP adipogenesis can improve muscle histopathology in vivo. 12Mo B6A/J mice were treated for 10 weeks with batimastat (2 mg/kg i. p. 3× weekly). As B6A/J gastrocnemius muscle showed significant adipogenic replacement starting from the age of 12Mo (Fig. 1i), we assessed for the extent of adipogenic loss in this muscle (Fig. 7c). Compared to the untreated mice, muscles from batimastat treated mice showed significantly reduced perilipin-1 labeled area (Fig. 7d). To determine if this reduction in adipogenic replacement was due to an effect on just FAP differentiation or also on FAP proliferation, we quantified FAP accumulation by PDGFRα labeling and saw no effect of batimastat treatment on FAP number (Supplementary Fig. 5).

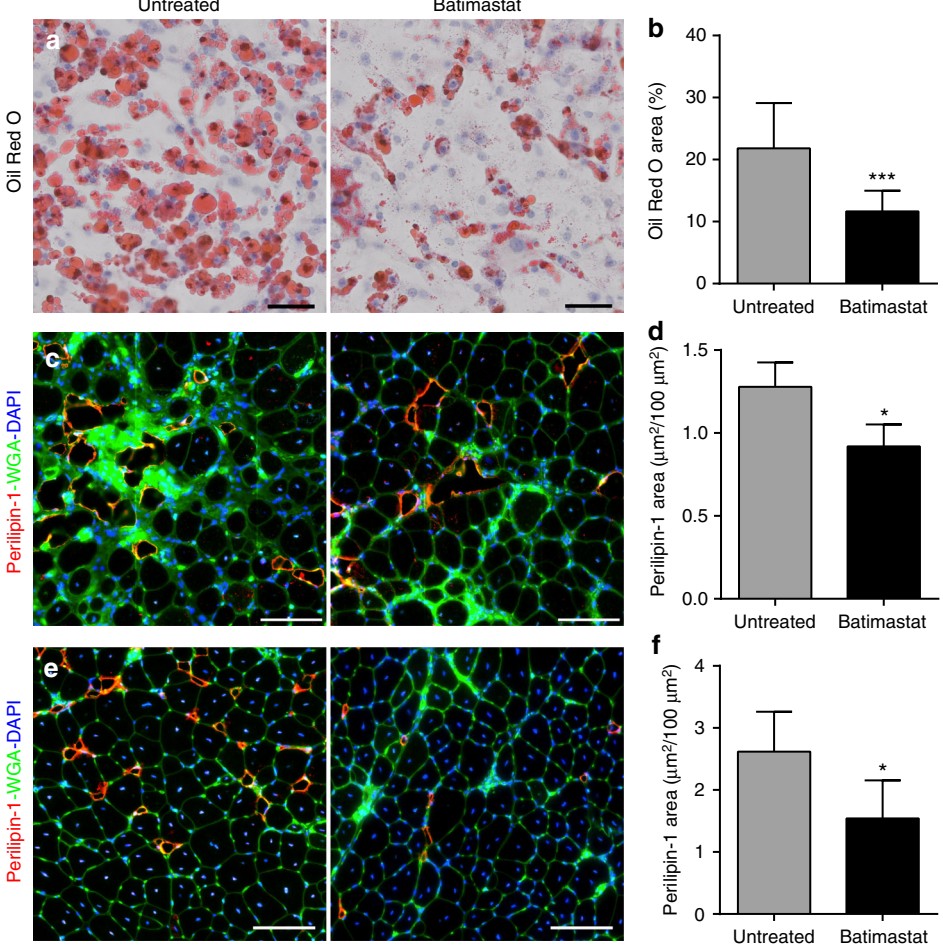

**Fig. 7** Batimastat blocks FAP adipogenesis and reduces adipogenic loss of dysferlinopathic muscle. **a** Oil Red O staining and **b** quantification (normalized to 20,000 FAPs) of spontaneous adipogenesis from 40,000 FAPs each, isolated from 12Mo B6A/J, after 10 days in culture (n = 3 replicates/group). Cells were treated with 10 μM batimastat starting day 3 in culture. Scale bar = 50 μm. **c** Perilipin-1 staining of 14Mo gastrocnemius muscle sections after 10 weeks of batimastat treatment. Scale bar = 100 μm. **d** Quantification of perilipin-1 stained area in gastrocnemius muscle, n = 3 muscles/group. **e** Perilipin-1 staining of TA muscles from 14Mo B6A/J mice after 3 successive notexin injuries while being treated (or left untreated) with batimastat. Scale bar = 100 μm. **f** Quantification of injury-triggered lipid formation in notexin injured TA muscles corresponding to panel (**e**), n = 6 muscles/group. All quantifications shown are mean ± SD, and statistical comparisons were performed via two-tailed t-test, *p < 0.05, ***p < 0.001

Similarly, batimastat treatment of 12Mo B6A/J mice did not decrease the extent of myofiber central nucleation (Supplementary Fig. 5). Thus, inhibition of adipogenesis by batimastat is not due to reduced FAP numbers or improved myofiber repair. To further confirm the ability of batimastat to prevent dysferlinopathic muscle from injury-triggered FAP adipogenesis, we employed the repeat notexin-induced TA injury approach which induces significant adipogenic replacement of the muscle (Fig. 3d, e). Compared to the untreated mice, TA muscles from batimastat treated mice showed a nearly 40% reduction in perilipin-1 positive area following repeat notexin injury (Fig. 7e, f). Again, this reduction was due to restriction of adipogenic differentiation of the FAPs and not due to an effect on their proliferation, as batimastat treatment did not reduce the FAP accumulation caused by repeat notexin injuries (Supplementary Fig. 5). Given that FAP numbers are unaffected by batimastat treatment, we also examined whether batimastat treatment causes cells to adopt a fibrogenic fate by blocking their adipogenic differentiation. However, we did not observe a change in intra-muscular fibrosis as a result of batimastat treatment in either the gastrocnemius or injured TA (Supplementary Fig. 5). Taken together, the above results both confirm the essential nature of FAP adipogenesis for the adipogenic conversion of dysferlinopathic muscle, and highlight the potential of batimastat to arrest the progressive and even late stage adipogenic replacement of dysferlinopathic muscle.

## Discussion

Myofiber loss and the associated muscle weakness is a feature of many muscular dystrophies. Thus, in the search for innovative therapies, it is important to elucidate the cellular mechanisms that link the initial genetic defect to disease onset and progression. In LGMD2B, the absence of dysferlin in myofibers inhibits sarcolemmal repair[21], disrupts proper calcium homeostasis at the t-tubules[27,41,42], and alters the response of innate immune cells[43,44]. Restoration of dysferlin expression in myogenic cells[45] and blockade of innate immune activation[46] reduce pathology in dysferlin-deficient mice, indicating that both the myofiber and inflammatory cell-specific deficits contribute to disease symptoms in dysferlinopathy. However, these deficits fail to explain the abrupt and late onset of disease in patients and the observation that adipogenic replacement is a feature of symptomatic dysferlinopathic muscle[28,29,35]. Here, we provide evidence that disease onset and progression in dysferlinopathy is not driven solely by the myofiber and inflammatory cell-specific defects, but by creation of an extracellular niche resulting in proliferation and adipogenic differentiation of muscle-resident FAPs.

From early in the disease, the primary myofiber defects lead to persistent myofiber damage. AnxA2, a dysferlin interacting protein that accumulates at the injured membrane and aids in its repair is released at the site of plasma membrane injury[47,48]. This AnxA2 released at the site of injured myofibers can trigger muscle inflammation, which in acute injury facilitates myofiber regeneration[49,50]. However, in LGMD2B patients AnxA2 levels are chronically increased in a manner that correlates with disease severity[31,32]. Given that AnxA2 expression is ubiquitous, AnxA2 can also be produced by myofibers or other cells present in the injured muscle, including the FAPs, endothelial and inflammatory cells. AnxA2 released chronically in the extracellular matrix of dysferlinopathic muscle creates a niche which favors increased proliferation and subsequently, adipogenic differentiation of FAPs (Fig. 8). Consistent with this hypothesis, we show that AnxA2 is a critical component of the pro-adipogenic FAP niche in dysferlinopathic muscle, as deletion of AnxA2 both decreases extracellular matrix FAP accumulation and prevents their

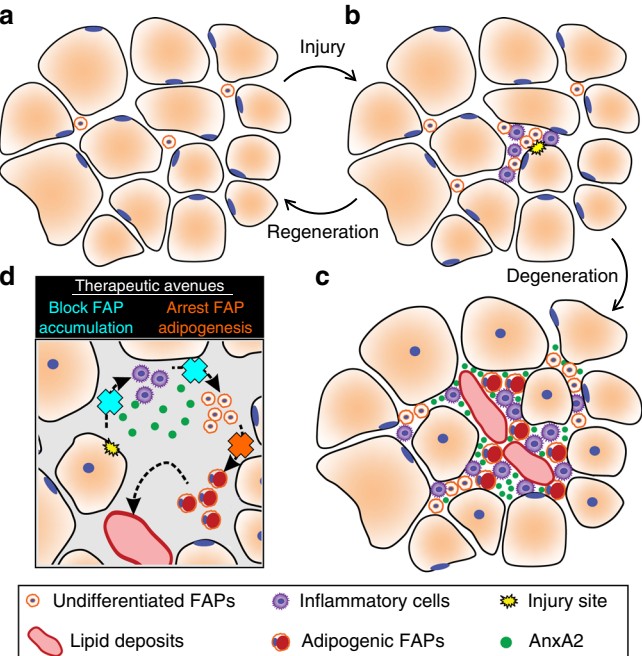

**Fig. 8** FAPs control the onset and severity of disease in LGMD2B. **a** Healthy and/or pre-symptomatic LGMD2B muscle contains resident FAPs. **b** After myofiber injury, inflammatory cells invade and trigger FAP proliferation. Successful regeneration involves a switch between pro-inflammatory and pro-regenerative signaling, causing the removal of inflammatory cells and FAPs. **c** In symptomatic LGMD2B muscle, there is a gradual accumulation of extracellular AnxA2, which prolongs the pro-inflammatory environment, causing excessive FAP proliferation. This cellular niche becomes pro-adipogenic over time, allowing for differentiation of FAPs and the adipogenic conversion of muscle. **d** Blocking aberrant signaling due to AnxA2 buildup blocks FAP accumulation and thus preventing adipogenic loss of dysferlinopathic muscle. Similarly, use of a MMP-14 inhibitor (Batimastat) inhibits FAP adipogenesis offering a potential drug-based therapy to prevent adipogenic loss of dysferlinopathic muscle

commitment to adipogenesis (Figs. 4 and 5). The presence of AnxA2 expressing niche cells in the muscle increases FAP adipogenesis and purified AnxA2 in regenerating muscle is capable of increasing FAP accumulation and adipogenesis in regenerating dysferlinopathic muscle (Figs. 6 and 7). These in vitro and in vivo analyses independently confirm an active role of AnxA2 in the adipogenic conversion of dysferlinopathic muscle.

In addition to the direct action of AnxA2 on FAPs, it may also act indirectly via inflammatory or other cell types. In support of such a role of AnxA2 we find that dysferlinopathic muscle lacking AnxA2 shows reduction in the extent of macrophage infiltration (Fig. 5). This could be due to the ability of AnxA2 to function as an agonist for innate immune response[38]. Previously, we showed that AnxA2 knockout in B6A/J mice down-regulates TLR4 signaling[33]. Inhibiting TLR signaling by deletion of the central TLR adapter protein Myd88 reduces pathology in dysferlinopathic (A/J) mice[46]. Innate immune signaling has been implicated in the dysferlinopathic symptoms[44], and dysferlin-deficient muscles show a greater abundance of pro-inflammatory as compared to pro-regenerative macrophages[51]. Together, these may contribute to delayed/impaired myogenesis[52], further enabling FAP accumulation[14]. Repeat rounds of myofiber injury, chronic inflammation, and FAP accumulation as the muscle ages may set up a feed-forward loop linking myofiber damage to the formation of a pro-adipogenic niche over time, which in turn contributes to myofiber damage (Fig. 8). In such a system,

adipogenic accumulation becomes the nucleating event that results in clinical onset, and an abrupt decline in muscle function in patients. Thus, while dysferlinopathy is driven by a myofiber-specific deficit, it is the impaired cellular interactions between myofibers, inflammatory cells, and FAPs that is causative for disease initiation and severity. This view of the disease opens up previously unrecognized avenues to intervene, as has been realized in DMD where inhibition of TGFβ-induced FAP accumulation reduces muscle fibrosis and leads to therapeutic benefits[19].

Use of *mdx* mice shows that aberrant FAP accumulation and differentiation drives disease pathogenesis in DMD[19]. While these FAPs primarily exhibit fibrogenic fate via TGFβ signaling, in vitro they also exhibit adipogenic potential[19,53]. Their adipogenic potential diminishes with age, advancing pathology, and by the use of HDAC inhibitors[53,54]. This contrasts with the in vivo increase in adipogenic fate of FAPs we observe in dysferlinopathic mice. This difference between the dystrophin (*mdx*) and dysferlin (B6A/J) mouse models suggests spontaneous fibrogenic commitment of the *mdx* and adipogenic commitment of B6A/J FAPs. However, the mechanism by which FAPs can choose between these fates remains elusive. Inhibition of MMPs is known to decrease the conversion of 3T3-L1 and primary rat preadipocytes into adipocytes[39]. Analyses of the FAPs show that pharmacological inhibition of matrix metalloprotease (MMP)-14 represses C/EBPδ and PPARγ in FAPs by way of cilial hedgehog signaling and this reduces the adipogenic fate of FAPs[40]. Adipogenesis of dysferlinopathic muscle has been linked with an increase in C/EBPδ and PPARγ mRNA in the muscle[28]. Both of these are essential transcription factors in adipocyte differentiation, and MMP-14 is suggested to be one of the extracellular signals that triggers adipogenic differentiation of FAPs. MMP-14 is released by myofibers and is critical to successful myogenesis during muscle regeneration[55]. Interestingly, MMP-14 expression in the dysferlin-null SJL mouse quadriceps increases by 3-fold between 2-month old (presymptomatic) and 9-month old (symptomatic) muscles[56]. Similar analysis of SJL mice has identified that the levels of AnxA1 and AnxA2 increase as these mice transition from 2 to 8 months of age[57]. Consistent with the putative role for MMP-14 in adipogenic conversion of dysferlinopathic muscle, we show that pharmacological inhibition of MMP-14 (by batimastat) reduces FAP adipogenesis in vitro and ameliorates injury-triggered lipid formation in vivo (Fig. 7). This suggests that gradual loss of FAP ciliation and/or repression of the Hh pathway may contribute to pro-adipogenic niche formation in dysferlinopathic muscle. Batimastat treatment has also been shown to reduce fibrosis and increase muscle function in *mdx* mice[58], which is suggestive of an anti-fibrotic effect on FAPs in dystrophin-deficient muscle. Because batimastat broadly inhibits MMPs, it is not clear whether this is due to specific inhibition of MMP-14 or other MMPs upregulated in *mdx* muscle[58]. But, it is clear that more insight into the mechanisms by which FAPs choose between fibrosis and adipogenesis and therapies targeting these pathways are of great interest in treating muscular dystrophies and other degenerative muscle diseases.

AnxA2-mediated MMP secretion has been shown to cause joint destruction in rheumatoid arthritis[59], suggesting AnxA2/MMP interactions may play a role in FAP-dependent adipogenesis in dysferlinopathic muscle. AnxA2 has also been shown to influence Hh signaling via the AnxA2 receptor in endothelial cells[58], providing additional mechanisms by which AnxA2 and MMP-14 may be linked during adipogenic niche formation in dysferlinopathic muscle. Such a role of AnxA2 is consistent with our previous study showing that loss of AnxA2 uncouples the repair defect of dysferlinopathic myofibers from the eventual adipogenic replacement of the muscle[33], identifying AnxA2 and MMP-14 as therapeutic targets for LGMD2B. This represents a

significant advance towards the development of the therapy for this disease, which currently lacks any effective or approved therapy. Unlike other inflammatory muscle diseases, where suppression of inflammation with corticosteroids is effective, treatment of dysferlinopathic patients with glucocorticoids has been without success[60], which may be due to the role of conventional corticosteroids in inducing myofiber damage and activating FAP adipogenesis[61,62]. Further, the use of tyrosine kinase inhibitors prevents excessive FAP proliferation in DMD mouse model[19,20], has toxicity associated with their long-term use required for LGMD2B. Our identification of inhibiting adipogenesis in dysferinopathic muscle by targeting FAPs by MMP-14 inhibitors (batimastat) also opens avenues for the use of other candidate drugs like promethazine, which also inhibits FAP adipogenesis[63]. In principle, the direct manipulation of PDGFRα splicing by morpholinos may also be beneficial by preventing FAP proliferation[64]. Irrespective of the precise therapeutic approach that would be efficacious, our study identifies the accumulation and adipogenic differentiation of FAPs as a central target to prevent the precipitation of cellular deficits into the abrupt onset of disease in dysferlinopathies. Moreover, such therapies would be complementary to the ongoing efforts to restore dysferlin expression in terminally differentiated myofibers.

## Methods

**Patient biopsies**. Patient biopsies were obtained under informed consent and was approved by the Ethics Committee of Hospital de la Santa Creu i Sant Pau de Barcelona. Frozen muscle biopsies from LGMD2B patients with 2 confirmed mutations in dysferlin were used. These were classified mild, moderate, and severe based on their clinical phenotype (Supplementary Table 1) for analysis. As a control, frozen muscle biopsies were obtained from young adults with no known neuromuscular conditions and without any histopathological features to serve as a comparison.

**Animals**. All animal procedures were conducted in accordance with guidelines for the care and use of laboratory animals, and were approved by the Children's National Medical Center Institutional Animal Care and Use Committee. C57BL/6J (WT) and B6.A-Dysf^prmd/GeneJ (B6A/J) mice were obtained from the Jackson Laboratory (Bar Harbor, ME) and maintained as homozygous colonies for the purpose of this study. A2-B6A/J mice were generated as part of our previous study[33], and are maintained in-house. All animals were maintained in an individually vented cage system under a controlled 12 h light/dark cycle with free access to food and water. Mice were used at the timepoints indicated in the study ± 2 weeks.

**Immunohistochemical analysis of muscle sections**. Frozen sections 8 μm thick were cut from human biopsies and the midbelly of mouse muscles. Lipid was visualized using an Oil Red O staining kit (American MasterTech, #KTORO) according to manufacturer's instructions. Immunofluorescence was performed by fixing sections in chilled 10% neutral buffered formalin, blocking with 5% BSA and incubation with primary antibodies against perilipin-1 (1:250, Sigma, #P1873), PDGFRα (1:250, Cell Signaling, #3174S), Annexin A2 (1:250, Santa Cruz, #SC-9061) and F4/80 (1:500, Serotec, #MCA497). For co-labeling anti-PDGFRα (1:100, R&D Systems, #AF1062) was used. Staining was visualized using relevant secondary antibodies conjugated to AlexaFluor 488 and/or 568 (1:500, Thermo-Fisher). Myofiber membranes were marked using AlexaFluor 488-conjugated wheat germ agglutinin (1:500, ThermoFisher, #W11261) and coverslips were mounted using ProLong Gold with DAPI (ThermoFisher, #P36941).

**Microscopy and image analysis**. Microscopy was performed using an Olympus BX61 VS120-S5 Virtual Slide Scanning System with UPlanSApo 40×/0.95 objective, Olympus XM10 monochrome camera or Allied vision Pike F-505C color camera, and Olympus VS-ASW FL 2.7 imaging software. Confocal images were acquired using an Olympus FV1000 Confocal Microscope with UPlanFLN 40×/1.30 oil objective and Olympus FV-ASW version 4.2 imaging software. Perilipin-1 quantification was performed by thresholding the image to exclude non-specific staining and then calculating the area of each lipid deposit encircled by perilipin-1 (excluding those in the epi- or perimysium) using MetaMorph software (Molecular Devices). The total area encircled by perilipin-1 was calculated for all lipid deposits across the muscle and expressed relative to the total cross-sectional area. PDGFRα and AnxA2 positive area was calculated using CellSens software (Olympus) by thresholding to remove non-specific staining and calculating the total positive area (again, excluding any epi- or perimysial staining) relative to the entire muscle cross-section.

**Isolation and in vitro adipogenesis of FAPs.** FAPs were isolated from the hindlimb muscles of 6Mo B6A/J mice in a modified protocol from previously published studies[8,10]. Mice were euthanized and the tibialis anterior, extensor digitorum longus, gastrocnemius, soleus, quadriceps, and psoas were immediately dissected. Non-muscle tissue including tendon, nerve, and overlying fascia were carefully removed, and muscles were minced finely in a sterile dish and incubated in Collagenase II (2.5 U/ml, ThermoFisher, #17101015) in PBS for 30 min at 37 °C. The resulting slurry was washed with sterile PBS before further digest in Collagenase D (1.5 U/ml, Sigma Aldridge, #COLLD-RO) and Dispase II (2.4 U/ml, Sigma Aldridge, #D4693) in PBS for 60 min at 37 °C. Resulting slurries were passed through 100 and 40 μm strainers and primary cells were resuspended in 1 ml PBS with 2% FBS and 2 nM EDTA. 300 μl of the primary cell suspension was reserved for plating and the remaining was stained with anti-PDGFRα-APC (1.0 μg per $10^6$ cells in 100 μl, Biolegend, #135908) and isotype control (1.0 μg per $10^6$ cells in 100 μl, Biolegend, #400512) for FACS. Cells stained with isotype control were used as a control for gating positive and negative events. Cells stained with anti-PDGFRα-APC were then sorted on an Influx cell sorter (Becton Dickinson, #646500) using two-way sorting to separate positive and negative PDGFRα-expressing cells for further analysis (Supplementary Fig. 6).

The freshly isolated primary cell, PDGFRα⁻ and PDGFRα⁺ populations were plated in Matrigel-coated Nunc Lab-Tek chamber slides (ThermoFisher, #154534) at a density of 20,000 cells/well. Cells were cultured in DMEM (Lonza, #12-604F) supplemented with 20% fetal bovine serum, 1% penicillin, and 2.5 ng/ml bFGF (BioLegend, #579604) for 3 days. Adipogenic differentiation was induced by exposure to DMEM with 10% FBS, 0.5 mM IBMX (Sigma Aldridge, #I5879), 0.25 μM dexamethasone (Sigma Aldridge, #D2915), and 10 μg/ml insulin (Sigma Aldridge, #I0516) for 3 days. Following this, cells were cultured in adipogenic maintenance media (DMEM with 10% FBS and 10 μg/ml insulin) for 3 days. Uninduced cells were not exposed to the adipogenic differentiation media, but instead cultured for 6 days in the adipogenic maintenance media, with the media changed after 3 days. At the beginning of day 10, cells were fixed for 30 min in chilled, neutral buffered formalin before Oil Red O staining to visualize lipid. For studies involving AnxA2 treatment, FAPs were isolated and plated as described above. From day 3 onwards, cells were continuously treated with 100 nM recombinant AnxA2 (RayBiotech, #230-30023) until being fixed for oil red staining on day 10.

**Notexin injury.** Single injury was performed by carefully shaving the anterior hindlimb before intramuscular injection of 40 μl notexin (5 μg/ml, Latoxan, #L8104) into the tibialis anterior using a 0.3 ml ultrafine insulin syringe (BD Biosciences, #324906). Immediately prior to injection, the needle was dipped in green tattoo dye (Harvard Apparatus, #72-9384) to mark the needle track. The contralateral leg was left uninjured as a control. Mice were allowed to recover for 28 days before the animal was euthanized and muscles collected for analysis. For repeat injuries we performed 3 separate intramuscular notexin injections, each 14 days apart, and allowed 28 days of recovery following the final injury before tissue collection. Using the superficial mark on the skin from the tattoo dye, we attempted to perform each injury as close to the site of the previous injury as possible so as to repeatedly injure the same myofibers each time. For AnxA2-notexin studies, 10 μg recombinant AnxA2 (RayBiotech, #230-30023) was added to 40 μl notexin (5 μg/ml, Latoxan, #L8104) and injected into the mid-belly of the right tibialis anterior. The contralateral (left) tibialis anterior was injected with 40 μl notexin only for comparison. Again, these muscles were harvested for analysis 28 days after injury.

**Batimastat treatment.** For in vitro studies, PDGFRα⁺ FAPs were isolated from 12Mo B6A/J muscle as described above, and plated in Matrigel-coated Nunc Lab-Tek chamber slides at 40,000 cells/well. Cells were cultured in DMEM (Lonza, #12-604F) supplemented with 20% fetal bovine serum, 1% penicillin, and 2.5 ng/ml bFGF (BioLegend, #579604) for 3 days. After which, cells were treated with 10 μM batimastat (Sigma-Aldridge, #SML0041) added to the adipogenic maintenance media for 6 days. At the beginning of day 10, cells were fixed for 30 min in chilled, neutral buffered formalin before Oil Red O staining to visualize lipid.

For in vivo studies, 12Mo B6A/J mice were treated thrice weekly with batimastat (Sigma-Aldridge, #SML0041), 2 mg/kg i.p. for 10 weeks. At the end of the treatment period, the effect of batimastat on disease pathology was evaluated by quantification of perilipin-1 marked lipid area in the gastrocnemius and compared to untreated controls. In addition, we tested injury-triggered adipogenesis using the repeat notexin injury protocol described above and in Supplementary Fig. 5. During the 10-week treatment period, 12Mo B6A/J mice were subjected to 3 intramuscular notexin injections into the TA, each 2 weeks apart, beginning on day 2 and ending on day 29. After 10 weeks of treatment, injury-induced lipid formation was quantified by perilipin-1 area in the injured TA and compared between batimastat treated and untreated controls. For both experiments, batimastat was first dissolved in DMSO, before being reconstituted in sterile 5% saline for treatment.

**Statistical analysis.** Data were analyzed using Prism GraphPad software. The precise statistical test employed varied depending on the nature of the analysis, and is listed in the legend for each figure. To visualize the distribution of data, we plotted the individual data points for each plot in Supplementary Figs. 7 and 8.

**Reporting summary.** Further information on research design is available in the Nature Research Reporting Summary linked to this article.

## Data availability
The authors declare that all data supporting the findings of this study are available within the paper and its Supplementary Information.

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

## Acknowledgements

This work is supported by a MDA Career Development Award (MDA477331) to M.W.H. Additional financial support was provided by NIAMS (R01AR055686) and MDA (MDA277389) to J.K.J. and NIH (K26OD011171; R24HD050846, P50AR060836) grants to K.N. The authors acknowledge Carsten Bönnemann's contribution to this study. Microscopy imaging was performed at the CRI Cellular Imaging Core, which is supported by funds from CRI and NICHD (U54HD090257).

## Author contributions

This study was conceptualized by M.W.H. and J.K.J., with contributions from K.N. and T.A.P. Experiments were designed by M.W.H. and J.K.J. and performed by M.W.H., with contributions from A.D. and C.L. Patient biopsies and clinical histories were collected and provided by E.G. and J.D.M. Data analysis was performed by M.W.H., with assistance from J.K.J. Funding for this project was obtained by M.W.H., J.K.J., and K.N. The initial manuscript was written by M.W.H. and J.K.J., and reviewed by T.A.P., K.N., C.L., and E.G. who all made contributions to the final manuscript text.
