## [Peer Review File · Nature Communications]

Reviewers' Comments:

Reviewer #1:

Remarks to the Author:

In this manuscript, the authors study the correlation between the extent of adipogenic infiltration and disease progression in a mouse model of dysferlinopathies. As FAP have been shown to be a key adipogenic cell type in skeletal muscle, they concentrate on these cells and they convincingly show, using mainly immunofluorescence and some in vitro experiments, that FAPs and adipogenic infiltration increases with damage and age, which are themselves correlated. Further, they show that deletion of AnxA2, a TLR ligand and a marker of membrane damage whose KO has been previously shown to ameliorate disease in this model, reduces adipogenesis and FAP increase as well. In general these data are solid, and they show that there is a strong correlation between muscle damage, number of FAPs present in the tissue and adipogenic infiltration. However, they do not support the interpretation, proposed multiple times in the manuscript, that the increase in FAPs and/or adipogenic infiltration directly causes disease progression. In fact, this relationship poses a classical "chicken or egg" problem that the authors make no attempt to address, leaving it completely unclear if the adipogenic infiltration is involved in pathogenesis or rather simply a marker of disease progression. This is clearly the case also in the AnxA2 KO experiments, as there is no attempt to understand if the change in the extent of adipogenic infiltration is directly caused by the lack of this gene or indirectly by the effects that the KO has on muscle damage.

This over-interpretation of causal linked that are merely suggested by the data is evident throughout the paper. For example, the authors show that, with time, there is an increase in FAPs in affected muscles and that FAPs are adipogenic in vitro, and they conclude that FAPs are the cause of adipogenic infiltration. While this conclusion is very likely to be correct, to formally establish that this is the case in vivo lineage tracing using an FAP-specific marker would be required.

As it is, this study is strictly descriptive and as such not fit for publication in this journal

Reviewer #2:

Remarks to the Author:

This paper by Hogarth et al describes that the number of FAPS seems to correlate with disease severity and adipogenic replacement in dysferlin deficiency. The authors have previously shown that deletion of Annexin A2 in dysferlin deficient mice results in decreased adipogenic replacement of fibers and improved muscle function despite worse membrane repair efficiency. Now the authors show that like other dystrophies that show fat replacement over time, that the number of FAPS in dysferlin deficient mice correlates with increased severity of disease and adipogenic replacement of muscle over lifespan. In mdx mice it has been shown that actually targeting FAPS for apoptotic clearance improves phenotype (the authors cite this work). While the data showing the correlation of FAP accumulation and disease progression in dysferlin deficiency is new, these interesting results stop short of definitively show cause and effect that the authors claim.

Major concerns:

1. While I agree that annexin A2 deletion in Bla/Jmice have less FAPS and fat accumulation, the double KO mice also have less disease severity and less inflammation. The authors have shown in the Figures 1-3 that severity is correlated with the number of FAPS, and influenced by the amount of prior injury and regeneration. Isn't it equally possible that annexin A2 regulates inflammation, which regulates severity and regeneration, and this then effects FAPS secondarily? I don't understand how cause and effect can be established by these observations in Figure 4. Indeed, the authors seem to speculate in the conclusion that the increase in FAPS could be also due to effects on inflammation
2. To definitively address whether A2/BlaJ mice have similar or altered adipogenic activity, the authors should isolate PDGFR cells from A2 KO and A2/BlaJ mice like in Figure 2 and test their

adipogenic capacity.

3. Notexin injuries may also affect myogenic cells and thus if there is competition between myogenesis and fatty accumulation after injury, the notexin may bias the injury toward adipogenic replacement with this non-physiological injury. Exercise has been shown to induce injury in dysferlin deficient mice. Does this accelerate adipogenic replacement? This would be important to know.

4. The authors conclude that annexin A2 is released into the ECM from injured muscles. Its not clear to me that annexin A2 is or isn't present on FAPS, or inflammatory cells or other resident cells in the muscle interstitium. Do the authors have data that annexin A2 is only expressed in muscle fibers?

Minor:

1. The title and short title overstate the conclusion. There is no data that demonstrates FAPs control or cause disease onset or severity in A/J mice or patients. There is only a correlation. Its possible Annexin A2 has other functions in muscle that effect onset and severity such as regulating inflammation, and that in turn regulates the number and/or activity of FAPS.

2. Figure 4 should show and compare F4/80 staining similar to data in Figure 4k.

3. The section describing the figure 3 results is confusing. There are typos in the text and the conditions in the results sections are not clear. The authors may wish to reorder the panels as well to make the important comparisons clearer. A WT mouse with repeated injury should be shown to compare to the repeated injury BlaJ mice.

Reviewer #3:

Remarks to the Author:

Hogarth et al. present evidence that FAP-mediated adipogenesis contributes to the pathogenesis of dysferlinopathies in a muscle, age, and annexin A2-dependent manner. The study is a follow-up to their recent report in Human Molecular Genetics and extends the dysferlinopathy/FAP story by establishing a correlation between the degree of FAP adipogenesis and the onset/progression of disease in both humans and mice. Importantly, the findings solidify FAP adipogenesis as a potential new therapeutic target to combat dysferlinopathy. Except as noted below, the manuscript was well written and easy to follow, appropriate controls were included, and proper statistics were employed. The work is of high importance for its clinical relevance to dysferlinopathy and should be of interest to anyone interested in membrane repair and/or muscular dystrophy.

Primary Concerns

1) An important conclusion in Fig 1 is that the perilipin-1 signal was between muscle fibers rather than within them. Unfortunately, the WGA/perilipin-1 images that support the claim are only reported in Sup Fig 1 and thus require the reader to compare Fig 1B to Sup Fig 1B to be convinced of the perilipin-1 localization. Including the WGA-data in Fig 1 would simplify interpretation of the data and improve the flow of the paper.

2) On Page 6 the authors state "In contrast, the regenerated TA showed a substantially more new adipogenic foci and myofiber areas replaced by adipogenesis, indicating age dependence of injury-triggered adipogenic replacement is age dependent (Fig. 3A - C)." In this sentence, the "regenerated TA" reference is ambiguous (does it refer to the 12 mo B6A/J single and double injury mice?) and the age-specific grammar is a mess. The authors should re-word the sentence to clarify and clean-up the mess.

3) A main conclusion from Fig 4 is that the level of adipogenesis is dramatically reduced in A2-B6A/J muscle. While the images presented in 4H and 4I are certainly convincing, they are simply single images; the figure would be much improved if the authors were to include a simple quantification of perilipin-1 signal similar to what they report in Fig 1E,I.

Secondary Concerns

- 1) On Page 4, bottom paragraph, in the reference (Fig. 1C, E, Supp. Fig. 1) the referral to Supp. Fig. 1 is not relevant as Sup Fig 1 does not include data from different muscle types as the referral suggests.
- 2) In the Fig 2 legend, "poathology" should instead read "pathology" and "mark the cells express" should read "mark the cells that express"

We thank the reviewers for their feedback appreciating the novelty and relevance of this work - "The work is of high importance for its clinical relevance to dysferlinopathy and should be of interest to anyone interested in membrane repair and/or muscular dystrophy." The additional data requested was to establish the cause and effect stated in the following comments by reviewers 1 and 2. The new work we have now conducted to address this and other issues have resulted in two new figures and inclusion of new data in three of the previous five figures. Below are pointwise responses to reviewer comments.

Reviewer 1

1. *In this manuscript, the authors study the correlation between the extent of adipogenic infiltration and disease progression in a mouse model of dysferlinopathies. As FAP have been shown to be a key adipogenic cell type in skeletal muscle, they concentrate on these cells and they convincingly show, using mainly immunofluorescence and some in vitro experiments, that FAPs and adipogenic infiltration increases with damage and age, which are themselves correlated. Further, they show that deletion of AnxA2, a TLR ligand and a marker of membrane damage whose KO has been previously shown to ameliorate disease in this model, reduces adipogenesis and FAP increase as well. In general these data are solid, and they show that there is a strong correlation between muscle damage, number of FAPs present in the tissue and adipogenic infiltration. However, they do not support the interpretation, proposed multiple times in the manuscript, that the increase in FAPs and/or adipogenic infiltration directly causes disease progression. In fact, this relationship poses a classical "chicken or egg" problem that the authors make no attempt to address, leaving it completely unclear if the adipogenic infiltration is involved in pathogenesis or rather simply a marker of disease progression. This is clearly the case also in the AnxA2 KO experiments, as there is no attempt to understand if the change in the extent of adipogenic infiltration is directly caused by the lack of this gene or indirectly by the effects that the KO has on muscle damage.*

Response: We and others previously showed that dysferlinopathies are late onset progressive disease with muscle weakness caused by gradual myofiber replacement with lipid (Defour et al. 2017; Grounds et al. 2014). This process is accelerated in patients by contraction-induced damage (Angelini et al. 2011). Based on these findings we had proposed that myofiber replacement with lipids causes muscle weakness in dysferlinopathy. However, pertinent to this comment, reviewer #2 suggested an alternate explanation for this wherein loss of dysferlinopathic myofiber may be caused by greater myofiber damage and tissue inflammation, and adipogenic replacement of muscle may be the end stage consequence (not cause) of myofiber loss. This suggestion captures the aforementioned 'chicken or egg' problem. The first clue against the alternate explanation comes from our previous finding that poor myofiber repair is common to both dysferlin-deficient (B6A/J) and dysferlin-Annexin A2 deficient (B6A/J-A2) mice, while adipogenic myofiber replacement occurs preferentially in B6A/J mice (Defour et al. 2017). This argues against differential muscle damage being the cause for differences between these mouse strains. Secondly, as use of glucocorticoid to prevent inflammation is ineffective in addressing the muscle weakness in LGMD2B patients (Angelini, Grisold, and Nigro 2011; Walter et al. 2013), it argues against the central role of inflammation in muscle loss. Our recent publication lends support to this claim where using B6A/J mice, we found that inhibiting inflammation by glucocorticoid (prednisone) treatment, fails to reduce weakness and adipogenic loss of dysferlin deficient muscle (Sreetama et al. 2018). Beyond these evidences contradicting the suggested alternate explanation, in the revised manuscript we provide following evidence to demonstrate our original hypothesis that FAP accumulation and differentiation is the underlying mechanism for myofiber loss in dysferlinopathy.

- A) Unlike the FAPs isolated from B6A/J muscle, FAPs isolated from A2-B6A/J muscle lack spontaneous adipogenesis (*Figure 5*), identifying that instead of reducing dysferlinopathic muscle damage,

Annexin A2 deficit prevents dysferlinopathic myofiber loss by acting directly at the level of prevention of FAP-dependent adipogenesis in this muscle.

- B) If adipogenesis is a consequence (not cause) of myofiber damage then drugs targeting FAP cannot prevent myofiber loss. Use of a drug that we found to inhibit adipogenic differentiation of dysferlinopathic FAPs *in vitro*, prevented spontaneous and injury-triggered adipogenic loss of dysferlinopathic muscle *in vivo* (Figure 6). This establishes FAP-dependent adipogenesis as the cause (not the consequence) of myofiber loss.

Above (published and new) findings establish the causal role of adipogenic differentiation of FAPs (not myofiber loss or muscle inflammation) for the muscle loss and weakness in dysferlinopathy.

Regarding if the change in the extent of adipogenic infiltration caused by Anx2KO is directly caused by the lack of this gene or indirectly by the effects that the KO has on muscle damage, we previously showed that AnxA2 KO causes poor myofiber repair and greater muscle fiber damage (indicated by high number of regenerated fibers (Defour et al. 2017)). This finding had demonstrated that deletion of AnxA2 on its own or in the context of dysferlin deficit does not reduce muscle damage. Instead what it does is protects the damaged muscle from undergoing adipogenic degeneration, the mechanism of which is what we have elucidated in this work (discussed above). In light of the above comment we realize the value of clarifying this matter and have provided additional data in the revised Figures 5 A – D, which show that while inflammation in A2-B6A/J muscle is reduced relative to B6A/J, it is still significantly elevated when compared to WT. This is in contrast to Perlipin-1 and PDGFR α levels, which are not different between A2-B6A/J and WT. These results offer evidence against adipogenic replacement in A2-B6A/J being an indirect consequence of excess muscle damage or inflammation.

2. *This over-interpretation of causal linked that are merely suggested by the data is evident throughout the paper. For example, the authors show that, with time, there is an increase in FAPs in affected muscles and that FAPs are adipogenic in vitro, and they conclude that FAPs are the cause of adipogenic infiltration. While this conclusion is very likely to be correct, to formally establish that this is the case in vivo lineage tracing using an FAP-specific marker would be required.*

Response: The initial parts of this comment (without the underline) are addressed in the responses above. Regarding the issue marked by the underlined text, we had shown in the original manuscript that the annexin A2-dysferlin double knockout (B6A/J-A2) mice have fewer FAPs and reduced adipogenic conversion of muscle. To establish that FAPs are the cause of adipogenic infiltration and thus the cause of degeneration of the dysferlin deficient muscle we first showed here that the FAPs isolated dysferlin deficient (B6A/J) muscle have increased spontaneous propensity to differentiate into adipocytes compared to FAPs from the B6A/J-A2 muscles (Figure 5G, H). Next, we show the efficacy of batimastat - a drug that inhibits adipogenic differentiation of wild type FAPs (Kopinke, Roberson, and Reiter 2017), to inhibit adipogenic fate of dysferlin deficient FAPs *in vitro* and reduce adipogenic infiltration and loss of dysferlinopathic muscle *in vivo* (Figure 6). This directly establishes the role of adipogenic differentiation of the FAPs in degeneration of dysferlinopathic muscle. We have thus gone beyond the lineage tracing to merely establish *in vivo* adipogenic potential of FAPs, and identified that targeting this will inhibit adipogenic loss of dysferlinopathic muscle.

3. *As it is, this study is strictly descriptive and as such not fit for publication in this journal*

As discussed above our findings that establish the adipogenic differentiation of FAPs as the mechanistic basis for the muscle loss and weakness in dysferlinopathy. Additionally, in resolving this issue we have identified a novel drug-based therapy to treat muscle loss in LGMD2B that targets FAP-mediated adipogenesis. This complements our previous finding that loss of Annexin A2 is a genetic therapy for

preventing adipogenic muscle loss in dysferlinopathy (*Figure 7*). Thus the work described in this manuscript goes well beyond description to establishing the mechanism and a potential drug based therapy to address this mechanistic deficit.

Reviewer 2

This paper by Hogarth et al describes that the number of FAPS seems to correlate with disease severity and adipogenic replacement in dysferlin deficiency. The authors have previously shown that deletion of Annexin A2 in dysferlin deficient mice results in decreased adipogenic replacement of fibers and improved muscle function despite worse membrane repair efficiency. Now the authors show that like other dystrophies that show fat replacement over time, that the number of FAPS in dysferlin deficient mice correlates with increased severity of disease and adipogenic replacement of muscle over lifespan. In mdx mice it has been shown that actually targeting FAPS for apoptotic clearance improves phenotype (the authors cite this work). While the data showing the correlation of FAP accumulation and disease progression in dysferlin deficiency is new, these interesting results stop short of definitively show cause and effect that the authors claim.

Major concerns:

- 1. While I agree that annexin A2 deletion in Bla/Jmice have less FAPS and fat accumulation, the double KO mice also have less disease severity and less inflammation. The authors have shown in the Figures 1-3 that severity is correlated with the number of FAPS, and influenced by the amount of prior injury and regeneration. Isn't it equally possible that annexin A2 regulates inflammation, which regulates severity and regeneration, and this then effects FAPS secondarily? I don't understand how cause and effect can be established by these observations in Figure 4. Indeed, the authors seem to speculate in the conclusion that the increase in FAPS could be also due to effects on inflammation*

Response: Please see the response to comment #1 of reviewer 1 on the issue of “cause and effect”. Regarding regulating adipogenesis by inhibiting inflammation, we would point to previous work in human showing the inefficacy of targeting muscle inflammation in preventing muscle weakness in LGMD2B patients (Angelini, Grisold, and Nigro 2011; Walter et al. 2013). We have now independently shown that merely inhibiting inflammation in dysferlinopathic mice does not correlate with reduced adipogenic muscle loss (Sreetama et al. 2018). Further, we have previously shown that the extent of myofiber regeneration in A2-B6A/J mice is comparable to B6A/J mice (Defour et al. 2017). Thus, if regeneration was the determining factor for FAP fate, then the FAP phenotype would not be any different between these mice. Our above results eliminate inflammatory and regenerative differences as the basis for the adipogenic loss of muscle in B6A/J mice and the lack of this in A2-B6A/J. While due to the effect of AnxA2 on TLR activation, inflammation may contribute to the tissue niche that facilitates adipogenic muscle loss in LGMD2B, our results identify AnxA2 effect on FAPs (*Figure 4, 5*) to be a more direct one and not a secondary outcome of inflammation or muscle regeneration.

- 2. To definitively address whether A2/BlaJ mice have similar or altered adipogenic activity, the authors should isolate PDGFR cells from A2 KO and A2/BlaJ mice like in Figure 2 and test their adipogenic capacity.*

Response: We agree with the reviewer regarding the need for this data and have conducted this experiment. This data shows that PDGFR α -labeled FAPs isolated from 12Mo B6A/J (symptomatic age for B6A/J) have significantly greater spontaneous adipogenic differentiation than FAPs isolated from matched A2-B6A/J and WT muscles (*Figures 5G, H*). This establishes that greater spontaneous adipogenic fate of B6A/J FAPs, which is not lost despite being maintained *in vitro* as mixed or purified cell culture.

3. *Notexin injuries may also affect myogenic cells and thus if there is competition between myogenesis and fatty accumulation after injury, the notexin may bias the injury toward adipogenic replacement with this non-physiological injury. Exercise has been shown to induce injury in dysferlin deficient mice. Does this accelerate adipogenic replacement? This would be important to know.*

Response: Previous work done in LGMD2B patients has also examined this idea and showed that injury caused by eccentric exercise exacerbates disease pathology and, pertinent to the reviewer's query, increases fatty replacement of muscle monitored by MRI (Angelini et al. 2011). As we mention in the introduction, this work was the premise for our experiments to test the effect of injury on muscle adipogenesis in B6A/J mice. Eccentric exercise has also been used to identify greater muscle damage in dysferlin-deficient mice (Biondi et al. 2013). However, in this study the authors did not examine adipogenic replacement of muscle. We took this approach and used downhill treadmill running to induce myofiber injury. However, in our hands eccentric injury induced by downhill treadmill run resulted in highly variable levels of myofiber injury (measured by induced central nucleation – see adjacent figure), making use of this injury approach unreliable to quantitatively assess the effect of exercise-induced injury on adipogenic response. However, as shown by the adjacent figure, extent of this injury (indicated by extent of centrally nucleated fibers) correlates positively with muscle adipogenic replacement (marked by perilipin-1 staining). This further supports that lipid formation is a physiological consequence of myofiber injury in dysferlinopathic muscle.

Lipid formation is positively correlated with myofiber injury after eccentric injury. 8Mo B6A/J mice were run downhill on a treadmill (12m/min, 30° decline) for 20min 3x weekly for 4 weeks to induce eccentric injury. Mice were left to recover for 4 weeks after the final bout of eccentric exercise before tissues were harvested for analysis. Lipid formation was quantified in the quadriceps via perilipin-1 staining and compared to myofiber central nucleation as an indicator of muscle damage. Despite wide variability in each of these measures as a result of the protocol, we observed a positive correlation between myofiber injury and lipid formation after eccentric injury.

Contrary to the reviewer's suggestion regarding notexin injury, a recent study showed that it is the glycerol-induced injury that impairs normal myogenic response and causes adipogenesis, which is avoided by the use of myotoxins (eg: notexin) where myogenic regeneration occurs efficiently within 14 days (Mahdy 2018). However, it is to control for any unanticipated effects of notexin injury on adipogenesis that we also tested adipogenesis in the muscles of asymptomatic mice - 12Mo WT and 3Mo B6A/J mice. This showed extensive regeneration at the injury site, indicated by the presence of centrally nucleated myofibers and a lack of necrotic myofibers, but no significant adipogenesis (*Figure 3A-F*), thus supporting the validity of using the notexin injury approach here. Finally, by isolating FAPs and culturing them *ex vivo* we show that the B6A/J FAPs activated by spontaneous *in vivo* injury adopt adipogenic fate (*figure 5G, H*). This independently shows that lipid deposits after notexin injury of the symptomatic 12Mo B6A/J is a reflection of the physiological fate of these muscles and not an artifact of the experimental approach.

4. *The authors conclude that annexin A2 is released into the ECM from injured muscles. Its not clear to me that annexin A2 is or isn't present on FAPS, or inflammatory cells or other resident cells in the muscle interstitium. Do the authors have data that annexin A2 is only expressed in muscle fibers?*

Response: Considering previous claims that Annexin A2 levels increase in the dysferlinopathic muscle (Cagliani et al. 2005) and the reports that AnxA2 is released at the site of plasma membrane injury

(Demonbreun et al. 2016; Jaiswal et al. 2014), our proposed model indicates release of AnxA2 by the injured muscle fibers. The released annexin molecules can then bind and reside on the surface of any of the cells in the vicinity including FAPs. AnxA2 can also be produced by other cell types (FAPs, endothelial and inflammatory cells) present in the injured muscle, and as long as these cells secrete AnxA2 these protein molecules would have the same effect as AnxA2 released by myofibers. Thus, in the 2nd paragraph of the revised discussion we have clarified these as potential sources for extracellular AnxA2 which can contribute to the adipogenic replacement of dysferlinopathic muscle.

Minor concerns

1. *The title and short title overstate the conclusion. There is no data that demonstrates FAPs control or cause disease onset or severity in A/J mice or patients. There is only a correlation. Its possible Annexin A2 has other functions in muscle that effect onset and severity such as regulating inflammation, and that in turn regulates the number and/or activity of FAPS.*

Response: Please see the responses above for the discussion of the causative role of FAPs in dysferlinopathic muscle adipogenesis. The revised *figure 7* summarizes the mechanisms identified in this study to target this role of FAPs *in vivo* to control disease severity in the mouse model. Regarding the role of Annexin A2 in regulation of inflammation, we fully agree with such a role of Annexin A2 and this is the working hypothesis of our ongoing work - please also see response to major comment #1 above.

2. *Figure 4 should show and compare F4/80 staining similar to data in Figure 4k.*

Response: In the revised *figure 5* we have provided the F4/80 staining images and a quantitative comparison of it in dysferlin deficient and dysferlin and AnxA2-deficient muscles.

3. *The section describing the Figure 3 results is confusing. There are typos in the text and the conditions in the results sections are not clear. The authors may wish to reorder the panels as well to make the important comparisons clearer. A WT mouse with repeated injury should be shown to compare to the repeated injury BlaJ mice.*

Response: We apologize for the poor wording used in describing the results presented in *figure 3*, leading to the confusion in the original manuscript. We have revised this text in this results section and in the figure legend to provide the required clarity.

Reviewer 3

Hogarth et al. present evidence that FAP-mediated adipogenesis contributes to the pathogenesis of dysferlinopathies in a muscle, age, and annexin A2-dependent manner. The study is a follow-up to their recent report in Human Molecular Genetics and extends the dysferlinopathy/FAP story by establishing a correlation between the degree of FAP adipogenesis and the onset/progression of disease in both humans and mice. Importantly, the findings solidify FAP adipogenesis as a potential new therapeutic target to combat dysferlinopathy. Except as noted below, the manuscript was well written and easy to follow, appropriate controls were included, and proper statistics were employed. The work is of high importance for its clinical relevance to dysferlinopathy and should be of interest to anyone interested in membrane repair and/or muscular dystrophy.

1. *An important conclusion in Fig 1 is that the perilipin-1 signal was between muscle fibers rather than within them. Unfortunately, the WGA/perilipin-1 images that support the claim are only reported in Sup Fig 1 and thus require the reader to compare Fig 1B to Sup Fig 1B to be*

convinced of the perilipin-1 localization. Including the WGA-data in Fig 1 would simplify interpretation of the data and improve the flow of the paper.

Response: We have revised figure 1, providing high resolution confocal images in which illustrate the point that perilipin-1 marked lipid deposits form outside myofibers in both, patient and mouse muscles.

- 2. On Page 6 the authors state “In contrast, the regenerated TA showed a substantially more new adipogenic foci and myofiber areas replaced by adipogenesis, indicating age dependence of injury-triggered adipogenic replacement is age dependent (Fig. 3A - C).” In this sentence, the “regenerated TA” reference is ambiguous (does it refer to the 12 mo B6A/J single and double injury mice?) and the age-specific grammar is a mess. The authors should re-word the sentence to clarify and clean-up the mess.*

Response: As mentioned above in our reply to Reviewer 2, point 8, we have made substantial revisions to the text in this section and the accompanying figure legend to provide clarity.

- 3. A main conclusion from Fig 4 is that the level of adipogenesis is dramatically reduced in A2-B6A/J muscle. While the images presented in 4H and 4I are certainly convincing, they are simply single images; the Figure would be much improved if the authors were to include a simple quantification of perilipin-1 signal similar to what they report in Fig 1E,I.*

Response: We have previously published a detailed account supporting this feature of A2-B6A/J muscle (Defour et al. 2017). Now in the revised manuscript we provide the images for reduced adipogenesis in gastrocnemius and quadriceps muscles and quantification corroborating this in the gastrocnemius (Figures 4H, I, 5C and Supplemental Figure 4).

- 4. On Page 4, bottom paragraph, in the reference (Fig. 1C, E, Supp. Fig. 1) the referral to Supp. Fig. 1 is not relevant as Sup Fig 1 does not include data from different muscle types as the referral suggests.*

Response: This has been corrected.

- 5. In the Fig 2 legend, “poathology” should instead read “pathology” and “mark the cells express” should read “mark the cells that express”*

Response: This has been corrected.

References

- Angelini, C, E Peterle, A Gaiani, L Bortolussi, and C Borsato. 2011. "Dysferlinopathy course and sportive activity: clues for possible treatment." *Acta Myologica* no. 30 (2):127.
- Angelini, Corrado, Wolfgang Grisold, and Vincenzo Nigro. 2011. "Diagnosis by protein analysis of dysferlinopathy in two patients mistaken as polymyositis." *Acta Myologica* no. 30 (3):185.
- Biondi, Olivier, Marie Villemeur, Alice Marchand, Fabrice Chretien, Nathalie Bourg, Romain K Gherardi, Isabelle Richard, and François-Jérôme Authier. 2013. "Dual effects of exercise in dysferlinopathy." *The American journal of pathology* no. 182 (6):2298-2309.
- Cagliani, Rachele, Francesca Magri, Antonio Toscano, Luciano Merlini, Francesco Fortunato, Costanza Lamperti, Carmelo Rodolico, Alessandro Prella, Manuela Sironi, and Mohammed Aguenouz. 2005. "Mutation finding in patients with dysferlin deficiency and role of the dysferlin interacting proteins annexin A1 and A2 in muscular dystrophies." *Human mutation* no. 26 (3):283-283.
- Defour, Aurelia, Sushma Medikayala, Jack H Van der Meulen, Marshall W Hogarth, Nicholas Holdreith, Apostolos Malatras, William Duddy, Jessica Boehler, Kanneboyina Nagaraju, and Jyoti K Jaiswal. 2017. "Annexin A2 links poor myofiber repair with inflammation and adipogenic replacement of the injured muscle." *Human molecular genetics* no. 26 (11):1979-1991.
- Demonbreun, Alexis R, Mattia Quattrocchi, David Y Barefield, Madison V Allen, Kaitlin E Swanson, and Elizabeth M McNally. 2016. "An actin-dependent annexin complex mediates plasma membrane repair in muscle." *J Cell Biol* no. 213 (6):705-718.
- Grounds, Miranda D, Jessica R Terrill, Hannah G Radley-Crabb, Terry Robertson, John Papadimitriou, Simone Spuler, and Tea Shavlakadze. 2014. "Lipid accumulation in dysferlin-deficient muscles." *The American journal of pathology* no. 184 (6):1668-1676.
- Jaiswal, Jyoti K, Stine P Lauritzen, Luana Scheffer, Masakiyo Sakaguchi, Jakob Bunkenborg, Sanford M Simon, Tuula Kallunki, Marja Jäättelä, and Jesper Nylandsted. 2014. "S100A11 is required for efficient plasma membrane repair and survival of invasive cancer cells." *Nature communications* no. 5.
- Kopinke, Daniel, Elle C Roberson, and Jeremy F Reiter. 2017. "Ciliary Hedgehog Signaling Restricts Injury-Induced Adipogenesis." *Cell* no. 170 (2):340-351. e12.
- Mahdy, Mohamed AA. 2018. "Glycerol-induced injury as a new model of muscle regeneration." *Cell and tissue research*:1-9.
- Sreetama, Sen Chandra, Goutam Chandra, Jack H Van der Meulen, Mohammad Mahad Ahmad, Peter Suzuki, Shivaprasad Bhuvanendran, Kanneboyina Nagaraju, Eric P Hoffman, and Jyoti K Jaiswal. 2018. "Membrane Stabilization by Modified Steroid Offers a Potential Therapy for Muscular Dystrophy Due to Dysferlin Deficit." *Molecular Therapy* no. 26 (9). doi: <https://doi.org/10.1016/j.ymthe.2018.07.021>.
- Walter, Maggie C, Peter Reilich, Simone Thiele, Joachim Schessl, Herbert Schreiber, Karlheinz Reiners, Wolfram Kress, Clemens Müller-Reible, Matthias Vorgerd, and Peter Urban. 2013. "Treatment of dysferlinopathy with deflazacort: a double-blind, placebo-controlled clinical trial." *Orphanet journal of rare diseases* no. 8 (1):26.

Reviewers' Comments:

Reviewer #1:

Remarks to the Author:

The authors have addressed my main concern with the following points:

1) FAPs from annexin A2 KO do not show spontaneous adipogenesis in vitro. This is a very interesting finding, but still does not establish a causal relationship. For example, it does not exclude the possibility that this may be due to a systemic effect altering FAP fate, which can't be excluded due to the fact that they are working with a developmental KO. The same effect could alter disease progression. Indeed the authors themselves claim that AnxA2 is expressed by myofibers, which are not present in the FAP cultures... Would coculturing AnxA2KO FAPs with myotubes from WT mice, or adding recombinant factor to the medium rescue the lack of adipogenesis?

2) Annexin A2 KO leads to a defect in muscle membrane repair, as does lack of dysferlin. The authors point to Defour et al 2017 in support of their contention that the effect on muscle fibers is similar, and that AnxA2 does not act by reducing such damage. However, their own histological analysis shows how AnxA2 muscle is essentially indistinguishable from the wild type, unlike dysferlin deficient muscle which clearly develops damage. The only in vivo evidence they bring forward supporting the notion that lack of AnxA2 causes damage is a faster decline in muscle strength that could be due to any of a number of mechanisms including some acting on nerve transmission or general metabolism.

3) A drug that alters adipogenesis also improves disease progression, which the authors take as proof of the involvement of adipogenic conversion of FAPs. But the new data shown in figure 6 does not establish a direct link, especially as the Bernatchez lab has recently been shown that in dysferlin deficient mice, alterations of fat metabolism have a profound effect on disease progression. In addition, the analysis of the outcome is fully focussed on adipogenesis and does not show if there is a benefit of the treatment in terms of muscle strength, muscle fiber size or any other indicator of disease progression apart from adipogenesis.

Finally, the authors show that Perlipin and PDGFR α levels are indistinguishable between A2-B6A/J and wild type. As PDGFR α is a marker of undifferentiated FAPs and is lost during adipogenic differentiation, this observation suggests that activation of FAPs may be impaired (which one would expect in the presence of less damage) rather than they differentiation, which would presumably lead to an increase in PDGFR α due to an increase in undifferentiated FAPs.

Importantly, a change in FAP numbers suggests that AnxA2 acts on more than their adipogenic propensity, as batimostat (the drug used by the authors to block adipogenesis) has been reported not to affect FAP numbers. Of course, other phenomena such as a secondary increase in apoptosis of undifferentiated FAPs may explain this observation, but the authors do not explore any alternative explanation in detail.

In conclusion, the results are still overinterpreted, with a number of experiments showing suggestive results that have not followed in enough depth to support the conclusions.

Reviewer #2:

Remarks to the Author:

The authors have written a careful rebuttal and addressed some of the criticisms raised in the previous review and also added some new data that have improved the manuscript. While I agree the collective data is supportive of the hypothesis that FAPS could be important in this phenotype, and the new data with the inhibitor adds to the supportive data, there are some concerns raised by the new data that if addressed would improve the paper.

1. The new data on a batimostat treatment provides some additional support, the data is somewhat inconclusive in terms of proving the mechanism. While BTMST inhibits FAP

differentiation in vitro my understanding is that it is a fairly broad MMP inhibitor and also inhibits things like angiogenesis and fibrosis. The authors should quantify the overall phenotype of the mice treated with drug, including the fibrosis, central nucleated fibers (as a marker of regeneration), and oil red O staining of the muscle itself (similar to Fig 1). The WGA staining shown, suggests there may be less fibrosis but it is not quantified. If the inhibitor also blocks fibrosis for example, the authors should temper their conclusions that this effect is mediated directly by the effects of BTMST on FAPS. For example, it could be working indirectly through affecting fibrosis. Alternatively, this data may suggest that inhibiting FAPS can significantly improve a dystrophic phenotype. Those two possible explanations could be discussed.

2. The new data on the FAPS isolated from mice show that the FAPS from B6A/J mice show heightened spontaneous adipogenic differentiation while the A2-B6A/J mice. This was a little surprising because the cells are isolated from the annexin secretory niche in the injured muscle that the authors are saying is driving their phenotype (although the authors argue they are already primed for differentiation). However, the experiment shown creates a unique opportunity to rescue the A2-B6A/J differentiation defect with recombinant A2 applied to the cells. This would definitively prove it is secreted A2 from injured cells that is priming this phenotype in strong support of their final model. Recombinant A2 is commercially available and it could be a great test of the model.

Minor

1. Fig 6B legend should indicate this is in the FAPS not in treated muscle.
2. Fig 6F legend should indicate if this is B6A/J mice from E just to be clear.
3. The methods should indicate if it was the same source of batimastat (Sigma) used in vivo and the carrier used in both the vitro and in vivo experiments. Sigma indicates solubility in DMSO. Was there a vehicle control in "untreated"?

Reviewer #3:

Remarks to the Author:

The Hogarth et al. revision is an improvement over the original submission with respect to both data presentation and readability. While my initial concerns were all addressed, minor issues remain in some figures and text (see below). However, in total the batimastat-related findings are significant and will appeal to those interested in MD in general and dysferlinopathy in particular.

Concerns

Text issues: while not an exhaustive list, the following are the most glaring mistakes.

- 1) On Page 7, the redundancy in "We next analyzed examined if AnxA2 ..." should be removed.
- 2) On Page 8, "adipogenic" is not a word
- 3) On Pg 10, "... inhibits MMPs, is not clear ..." should read "... inhibits MMPs, it is not clear ..."

Figure issues

Fig. 1A-B: no scale bar is provided

Supp Fig 1: no "C" is included, in disagreement with the legend.

We thank the reviewers for their positive comments and for their enthusiasm and feedback to further enhance the impact of our work. Below we have included a point wise response to their comments presenting our new experimental results and other edits to the manuscript that addresses each of the points they have raised.

Reviewer #1

The authors have addressed my main concern with the following points:

1) FAPs from annexin A2 KO do not show spontaneous adipogenesis in vitro. This is a very interesting finding, but still does not establish a causal relationship. For example, it does not exclude the possibility that this may be due to a systemic effect altering FAP fate, which can't be excluded due to the fact that they are working with a developmental KO. The same effect could alter disease progression. Indeed the authors themselves claim that AnxA2 is expressed by myofibers, which are not present in the FAP cultures... Would coculturing AnxA2KO FAPs with myotubes from WT mice, or adding recombinant factor to the medium rescue the lack of adipogenesis?

Response: This comment raises two interesting questions - 1. Is the adipogenic ability of AnxA2 KO FAPs compromised due to the developmental KO of AnxA2? 2. Does AnxA2 released by myofibers and other cells enhance FAP adipogenesis? We have carried out experiments that address each of these concerns.

1. To address the adipogenic ability of FAPs lacking AnxA2 we tackled the more challenging model AnxA2/dysferlin dKO, where lack of AnxA2 reverses the increase in FAP adipogenesis caused by dysferlin deficit. Treating these FAPs with media used for in vitro adipogenic induction we find that these cells robustly adopt the adipogenic fate (Fig. S4C, D), producing lipids detectable by oil red staining. This establishes that AnxA2 KO does not prevent the developmental potential of FAPs to adopt the adipogenic fate.
2. To assess the extracellular role of AnxA2 in FAP adipogenesis we first examined if purified AnxA2 can further enhance the adipogenic fate of the FAPs. We find that treating the B6A/J FAPs with recombinant AnxA2 further increased their adipogenesis, directly addressing the role of extracellular AnxA2 in augmenting FAP adipogenesis (Fig. 6E, F). In vivo, extracellular AnxA2 likely arises from multiple different types of cells in the damaged muscle niche. So we examined if lack of AnxA2 in the niche cells from dKO muscle inhibit adipogenesis. For this we co-cultured B6A/J FAPs with the niche cells from either B6A/J or A2-B6A/J. Interestingly, compared to the B6A/J niche, co-culture with the A2-B6A/J niche reduced spontaneous adipogenesis of B6A/J FAPs (Fig. 6C, D). These opposing effects of the presence/absence of extracellular AnxA2 demonstrates the AnxA2 secretion by the cellular niche surrounding the FAPs facilitates FAP adipogenesis.
3. Finally, we tested the potential of extracellular AnxA2 to induce FAP adipogenesis *in vivo*. For this we performed notexin injury in A2-B6A/J muscle and injected recombinant AnxA2 to raise

the extracellular AnxA2 level in these muscles during the course of muscle regeneration; mimicking the high AnxA2 in the extracellular niche found in the B6A/J mice. Mere presence of exogenous AnxA2 during regeneration of these otherwise AnxA2 deficient dysferlinopathic muscle, increased both FAP accumulation and adipogenesis during the course of muscle regeneration (Fig. 6G – J). Thus, providing *in vivo* confirmation that it is the presence of extracellular AnxA2 that drives the adipogenic niche for FAPs in the dysferlinopathic muscle.

2) Annexin A2 KO leads to a defect in muscle membrane repair, as does lack of dysferlin. The authors point to Defour et al 2017 in support of their contention that the effect on muscle fibers is similar, and that AnxA2 does not act by reducing such damage. However, their own histological analysis shows how AnxA2 muscle is essentially indistinguishable from the wild type, unlike dysferlin deficient muscle which clearly develops damage. The only in vivo evidence they bring forward supporting the notion that lack of AnxA2 causes damage is a faster decline in muscle strength that could be due to any of a number of mechanisms including some acting on nerve transmission or general metabolism.

Response: In our previous manuscript (Defour et al., 2017) we had employed two assays (focal laser injury and mechanical injury by ex vivo lengthening contractions) that are the field's standard for directly assessing muscle membrane repair ability independent of potential effects due to factors such as nerve transmission and metabolic or behavioral deficits. This showed that WT mice knocked out for AnxA2 show slower muscle membrane repair (Fig 1A-D and Fig. 4C in Defour et al., 2017). Even more pertinent to our current manuscript was the findings that knockout of ANxA2 in the dysferlin-deficient mice resulted in further reduction of the muscle membrane repair ability (Fig. 4D and 6C in Defour et al., 2017). These previous observations we made regarding poor muscle membrane repair sufficiently explains what is noted above by the reviewer regarding weakness of AnxA2 deficient muscle. Further, not only has the past research with AnxA2 knockout mouse by other laboratories not identified a general nerve transmission or related deficit in this mouse, our own independent work has demonstrated that metabolic poisoning of cells with azide (mitochondrial poison) and 2-deoxyglucose (glycolytic poison) does not impact on muscle membrane repair ability (Horn et al., 2017). This in light of this body of data in favor of poor membrane repair deficit and against other potential deficits, our finding that "*Annexin A2 KO leads to a defect in muscle membrane repair, as does lack of dysferlin.*" is borne out by all of the existing data and is not a "contention". Our findings noted above by the reviewer regarding lack of muscle histopathology in the AnxA2 deficient muscles is what prompted us to ask the question in the previous (Defour et al., 2017) as well as in the current study about what factor(s) beyond muscle membrane repair contribute to the histopathological damage seen in dysferlin deficient, but not in AnxA2 deficient muscles. The findings we present in this current study helps answer this by highlighting that it is the breakdown in the interactions between the poor repairing muscle fiber and the surrounding niche made up of non-muscle (FAPs and inflammatory) cells that cause the ensuing histopathology in dysferlin deficient, but not in AnxA2 deficient, muscle.

3) A drug that alters adipogenesis also improves disease progression, which the authors take as proof of the involvement of adipogenic conversion of FAPs. But the new data shown in figure 6 does not establish

a direct link, especially as the Bernatchez lab has recently been shown that in dysferlin deficient mice, alterations of fat metabolism have a profound effect on disease progression. In addition, the analysis of the outcome is fully focussed on adipogenesis and does not show if there is a benefit of the treatment in terms of muscle strength, muscle fiber size or any other indicator of disease progression apart from adipogenesis.

Response: Batimastat - the drug we have used here is a MMP14 inhibitor (Hotary et al., 2002) and inhibits the signaling required to trigger adipogenic fate of FAPs (Kopinke et al., 2017). In the dysferlin-null mouse, MMP-14 expression increases by 3-fold between the ages of 2 months old (presymptomatic) to 9 months old (symptomatic) (Suzuki et al., 2005). Thus ability of batimastat to reduce adipogenesis of isolated FAPs *in vitro* and *in vivo* in symptomatic dysferlin KO muscle identifies FAPs as the target for reducing adipogenic muscle loss in dysferlinopathy. This is also supported by batimastat reversing the accelerated adipogenic loss of dysferlin KO muscles caused by FAP activation by notexin injury (Fig. 7E, F). None of our findings are in context of the metabolic stress imposed on the mice by the Bernatchez lab through loss of ApoE protein and maintaining mice on high fat diet - treatments that trigger metabolic dysfunction leading to major atherosclerotic lesions in these mice (Plump et al., 1992). This same metabolic stress also causes adipogenic replacement of muscles even in *mdx* mice (Milad et al., 2017). It thus appears that ApoE and altered diet-induced muscle adipogenic loss is independent of the dystrophy model and does not reflect upon a specific endogenous metabolic alteration in dysferlin KO. Finally, the short-term batimastat treatment presented in the manuscript is a proof of principle demonstration of the relevance of FAP adipogenesis to the adipogenic loss of dysferlin-deficient muscle. Measurement of functional muscle improvements in a controlled and properly powered preclinical assessment needs to be performed, which is clearly beyond the scope of this study aimed at describing the mechanism and proof-of-concept for such a preclinical trial in the future.

4) The authors show that Perlipin and PDGFR α levels are indistinguishable between A2-B6A/J and wild type. As PDGFR α is a marker of undifferentiated FAPs and is lost during adipogenic differentiation, this observation suggests that activation of FAPs may be impaired (which one would expect in the presence of less damage) rather than they differentiation, which would presumably lead to an increase in PDGFR α due to an increase in undifferentiated FAPs. Importantly, a change in FAP numbers suggests that AnxA2 acts on more than their adipogenic propensity, as batimostat (the drug used by the authors to block adipogenesis) has been reported not to affect FAP numbers. Of course, other phenomena such as a secondary increase in apoptosis of undifferentiated FAPs may explain this observation, but the authors do not explore any alternative explanation in detail.

Response: Our *in vitro* experiments described in response to comment #1 above directly demonstrates the role of AnxA2 in adipogenic differentiation of the FAPs. This complements the data showing reduced adipogenic differentiation ability of the AnxA2 deficient FAPs (Fig. 6A, B) contributes to the reduced adipogenic fate of the A2-B6A/J muscles. Our new experiment where we injected AnxA2-B6A/J mouse muscles with purified AnxA2 during the course of regeneration showed that presence of AnxA2 results not only in increased FAP adipogenesis, but also in increased number of FAPs. This *in vivo* evidence adds further to the role of AnxA2 also in the accumulation of FAPs. Thus, as the reviewer suggests we believe

that absence of AnxA2 in the knockout mice may reduce the number and adipogenic differentiation of FAPs, leading to the lack of FAP and adipogenic accumulation in these mice. In the spirit of seeking other explanation and in response to the comment # 1 of reviewer 2, we have tested another potential alternative effect of AnxA2 on FAPs – switch from adipogenic to fibrotic differentiation. As we described in that response below, the evidence we obtained points against this alternate explanation.

In conclusion, the results are still overinterpreted, with a number of experiments showing suggestive results that have not followed in enough depth to support the conclusions.

Response: The new data we have presented in the revised manuscript addresses each of the above comments of this reviewer.

Reviewer #2

The authors have written a careful rebuttal and addressed some of the criticisms raised in the previous review and also added some new data that have improved the manuscript. While I agree the collective data is supportive of the hypothesis that FAPS could be important in this phenotype, and the new data with the inhibitor adds to the supportive data, there are some concerns raised by the new data that if addressed would improve the paper.

1) The new data on a batimastat treatment provides some additional support, the data is somewhat inconclusive in terms of proving the mechanism. While BTMST inhibits FAP differentiation in vitro my understanding is that it is a fairly broad MMP inhibitor and also inhibits things like angiogenesis and fibrosis. The authors should quantify the overall phenotype of the mice treated with drug, including the fibrosis, central nucleated fibers (as a marker of regeneration), and oil red O staining of the muscle itself (similar to Fig 1). The WGA staining shown, suggests there may be less fibrosis but it is not quantified. If the inhibitor also blocks fibrosis for example, the authors should temper their conclusions that this effect is mediated directly by the effects of BTMST on FAPS. For example, it could be working indirectly through affecting fibrosis. Alternatively, this data may suggest that inhibiting FAPS can significantly improve a dystrophic phenotype. Those two possible explanations could be discussed.

Response: We agree with the reviewer's suggestion that, despite the use of batimastat to inhibit MMP-14, it may also inhibit other MMPs and we have thus quantified the other phenotypic parameters recommended by the reviewer. While fibrosis is not a major feature of dysferlinopathy, we performed further analysis of batimastat-treated mouse muscles to examine if the reduced lipid formation is caused by a switch to fibrosis and/or myofiber damage rather than reduced FAP adipogenesis. Masson's trichrome staining, which is a more specific indicator of fibrosis than WGA, showed no effect of batimastat on muscle fibrosis, either with spontaneous pathology in the gastrocnemius or with notexin-induced injury in the TA (Fig. S5F - I). We also quantified central nucleation in the gastrocnemius and found no significant difference between batimastat –treated and untreated B6A/J muscle (Fig. S5C). This new data, coupled with our previous observation that batimastat reduces FAP adipogenesis *in vitro*, demonstrates that the reduction in lipid area in batimastat treated mice is due to a reduction in FAP adipogenesis as we had previously suggested.

2) The new data on the FAPS isolated from mice show that the FAPS from B6A/J mice show heightened spontaneous adipogenic differentiation while the A2-B6A/J mice. This was a little surprising because the cells are isolated from the annexin secretory niche in the injured muscle that the authors are saying is driving their phenotype (although the authors argue they are already primed for differentiation). However, the experiment shown creates a unique opportunity to rescue the A2-B6A/J differentiation defect with recombinant A2 applied to the cells. This would definitively prove it is secreted A2 from injured cells that is priming this phenotype in strong support of their final model. Recombinant A2 is commercially available and it could be a great test of the model.

Response: We have performed new experiments to address this suggestion by the reviewer. First, we tested the idea that the cells surrounding the FAPs (niche cells) regulate the adipogenesis of dysferlinopathic FAPs. For this we co-cultured the B6A/J FAPs with niche cells from either B6A/J or age-matched A2-B6A/J. In this scenario, the A2-B6A/J niche cells reduced the adipogenesis of B6A/J FAPs (Fig. 6C, D). Next, we tested whether this niche effect was a consequence of the lack of AnxA2 supplied by the A2-B6A/J niche. For this we treated FAPs extracted from B6A/J with recombinant AnxA2 and observed a significant increase in their spontaneous adipogenic potential compared with untreated cells (Fig. 6E, F). This suggests a direct action by AnxA2 on FAPs, which is in line with our previous observations that the increase in AnxA2 coincides with the accumulation of FAPs in dysferlin-deficient muscle. Finally, we tested whether this *in vitro* effect of AnxA2 also applies *in vivo* by notexin injuring AnxA2-naïve A2-B6A/J muscle in the presence or absence of purified AnxA2. In this experiment, the exogenous AnxA2 increased both injury-triggered FAP accumulation and adipogenic replacement of A2-B6A/J muscle (Fig. 6G – J), thus confirming the pro-adipogenic action of AnxA2 on dysferlinopathic FAPs.

Minor

1. Fig 6B legend should indicate this is in the FAPs not in treated muscle.

Response: The figure legend (now Fig. 7B) has been updated to be explicitly clear the Oil Red quantification refers to 12Mo B6A/J FAPs *in vitro*.

2. Fig 6F legend should indicate if this is B6A/J mice from E just to be clear.

Response: The figure legend for this figure (now 7F) has been edited to be clear this refers to the NTX-injured B6A/J muscles shown in the adjacent panel 7E.

3. The methods should indicate if it was the same source of batimastat (Sigma) used *in vivo* and the carrier used in both the *in vitro* and *in vivo* experiments. Sigma indicates solubility in DMSO. Was there a vehicle control in “untreated”?

Response: We apologize for this admission in the methods. The source of batimastat was the same for both *in vitro* and *in vivo* experiments. The drug was first dissolved in DMSO before being reconstituted in 5% saline for treatment. The methods have been updated to reflect this detail.

Reviewer #3

The Hogarth et al. revision is an improvement over the original submission with respect to both data presentation and readability. While my initial concerns were all addressed, minor issues remain in some figures and text (see below). However, in total the batimastat-related findings are significant and will appeal to those interested in MD in general and dysferlinopathy in particular.

Concerns

Text issues: while not an exhaustive list, the following are the most glaring mistakes.

1) On Page 7, the redundancy in “We next analyzed examined if AnxA2 ...” should be removed.

Response: This has been corrected, and the word ‘analyzed’ was removed from the sentence.

2) On Page 8, “adipogenic” is not a word

Response: This typo has been corrected.

3) On Pg 10, “... inhibits MMPs, is not clear ...” should read “... inhibits MMPs, it is not clear ...”

Response: The sentence has been changed as suggested.

Figure issues

Fig. 1A-B: no scale bar is provided

Response: Scale bar has been provided for these images.

Supp Fig 1: no “C” is included, in disagreement with the legend.

Response: The numbering of this figure has been corrected.

References

Defour, A., Medikayala, S., Van der Meulen, J.H., Hogarth, M.W., Holdreith, N., Malatras, A., Duddy, W., Boehler, J., Nagaraju, K., and Jaiswal, J.K. (2017). Annexin A2 links poor myofiber repair with inflammation and adipogenic replacement of the injured muscle. *Human molecular genetics* 26, 1979-1991.

Horn, A., Van der Meulen, J.H., Defour, A., Hogarth, M., Sreetama, S.C., Reed, A., Scheffer, L., Chandel, N.S., and Jaiswal, J.K. (2017). Mitochondrial redox signaling enables repair of injured skeletal muscle cells. *Sci Signal* 10, eaaj1978.

Hotary, K.B., Yana, I., Sabeh, F., Li, X.-Y., Holmbeck, K., Birkedal-Hansen, H., Allen, E.D., Hiraoka, N., and Weiss, S.J. (2002). Matrix Metalloproteinases (MMPs) Regulate Fibrin-invasive Activity via MT1-MMP–dependent and –independent Processes. *The Journal of Experimental Medicine* 195, 295-308.

Kopinke, D., Roberson, E.C., and Reiter, J.F. (2017). Ciliary Hedgehog Signaling Restricts Injury-Induced Adipogenesis. *Cell* 170, 340-351. e312.

Milad, N., White, Z., Tehrani, A.Y., Sellers, S., Rossi, F.M., and Bernatchez, P. (2017). Increased plasma lipid levels exacerbate muscle pathology in the mdx mouse model of Duchenne muscular dystrophy. *Skeletal muscle* 7, 19.

Plump, A.S., Smith, J.D., Hayek, T., Aalto-Setälä, K., Walsh, A., Verstuyft, J.G., Rubin, E.M., and Breslow, J.L. (1992). Severe hypercholesterolemia and atherosclerosis in apolipoprotein E-deficient mice created by homologous recombination in ES cells. *Cell* 71, 343-353.

Suzuki, N., Aoki, M., Hinuma, Y., Takahashi, T., Onodera, Y., Ishigaki, A., Kato, M., Warita, H., Tateyama, M., and Itoyama, Y. (2005). Expression profiling with progression of dystrophic change in dysferlin-deficient mice (SJL). *Neuroscience research* 52, 47-60.

Reviewers' Comments:

Reviewer #1:

Remarks to the Author:

The authors have made significant efforts to address my concerns. While I was somewhat disappointed that they did not assess the effects of batimastat treatment on muscle function, the paper is now publishable.

Reviewer #2:

Remarks to the Author:

The author addressed my concerns.

This is not a criticism that needs to be addressed but I did notice that taking the experiments as a whole, the perilipin staining appears to be concentrated around the smaller caliber fibers that remain after notexin injury. It might be interesting to mention this and perhaps follow this up why this non-uniform distribution exists in future experiments.

REVIEWERS' COMMENTS:

Reviewer #1 (Remarks to the Author):

The authors have made significant efforts to address my concerns. While I was somewhat disappointed that they did not assess the effects of batimastat treatment on muscle function, the paper is now publishable.

Response: We wish to thank the reviewer for their efforts in improving this manuscript over the course of review. We agree that the effect of long term batimastat treatment on disease progression, including muscle function, FAP accumulation and adipogenic replacement in B6A/J mice is of great interest. As such, conducting a rigorous preclinical trial in this model is a priority in our work moving forward.

Reviewer #2 (Remarks to the Author):

The author addressed my concerns.

This is not a criticism that needs to be addressed but I did notice that taking the experiments as a whole, the perilipin staining appears to be concentrated around the smaller caliber fibers that remain after notexin injury. It might be interesting to mention this and perhaps follow this up why this non-uniform distribution exists in future experiments.

Response: The areas referred to here (Figure 3) show Perilipin accumulation in the extracellular matrix spaces between the myofibers and not around (surrounding) intact small caliber myofibers. This is evident by lack of any nuclei within these spaces surrounded by the accumulating lipids. To avoid this confusion for the readers we have now reworded the relevant result section as follows: "This showed that increasing age increases injury-triggered adipogenic replacement of areas otherwise occupied by myofibers."